# A single 2′-*O*-methylation of ribosomal RNA gates assembly of a functional ribosome

James N. Yelland [1], Jack P. K. Bravo[2], Joshua J. Black [2,5],
David W. Taylor [1,2,3,4] ✉ & Arlen W. Johnson [1,2] ✉

RNA modifications are widespread in biology and abundant in ribosomal RNA. However, the importance of these modifications is not well understood. We show that methylation of a single nucleotide, in the catalytic center of the large subunit, gates ribosome assembly. Massively parallel mutational scanning of the essential nuclear GTPase Nog2 identified important interactions with rRNA, particularly with the 2′-*O*-methylated A-site base Gm2922. We found that methylation of G2922 is needed for assembly and efficient nuclear export of the large subunit. Critically, we identified single amino acid changes in Nog2 that completely bypass dependence on G2922 methylation and used cryoelectron microscopy to directly visualize how methylation flips Gm2922 into the active site channel of Nog2. This work demonstrates that a single RNA modification is a critical checkpoint in ribosome biogenesis, suggesting that such modifications can play an important role in regulation and assembly of macromolecular machines.

Chemical modification of RNA is pervasive in all three domains of life and is abundant in rRNA. Most rRNA modifications cluster around functionally important regions of the ribosome, including the decoding center and the peptidyl transferase center, where they are thought to stabilize folding and the tertiary structure of RNA at these functionally important sites[1–3]. A noteworthy example is the universally conserved A-site loop of rRNA helix 92 (H92), which is highly modified and contains the 2′-*O*-methylated guanosine (Gm2922 in yeast). During translation, Gm2922 base-pairs with the 3′-end of an accommodated transfer RNA to position it in the A-site for the peptidyl transferase reaction[4]. However, the importance of G2922 methylation, and indeed of any specific rRNA modification, remains poorly understood.

Eukaryotic ribosome assembly is a complex and intricately organized pathway spanning multiple cellular compartments[5,6]. Over 200 assembly factors play a role in assembling the functional ribosome, including several evolutionarily conserved ATPases and GTPases which drive rRNA rearrangements and progression of ribosome assembly. One such enzyme is Nog2, an essential nuclear GTPase conserved

between yeast and humans. Nog2 is required during critical RNA rearrangements within the precursor large (pre-60S) subunit, before nuclear export of the subunit[7]. However, the specific function of Nog2 has not been identified. To better understand the function of this GTPase, we searched for functionally important regions of yeast Nog2 using massively parallel mutagenic scanning. Unexpectedly, we discovered that physical interaction of Nog2 with H92, including recognition of the 2′-*O*-methylated base Gm2922, is critical for cell viability. We found that pre-60S subunits lacking methylated G2922 are blocked for assembly and nuclear export, and that orthogonal 2′-*O*-methylation of G2922 or single amino acid changes in Nog2 overcame the ribosome assembly and nuclear export defects of pre-60S lacking methylated G2922. Finally, we used cryoelectron microscopy (cryo-EM) to solve the structure of a nascent 60S subunit from cells with unmethylated G2922, showing that an unmethylated G2922 fails to engage with Nog2. The present study thus demonstrates an important role for a single rRNA modification as a structural checkpoint in the eukaryotic ribosome assembly pathway.

[1]Interdisciplinary Life Sciences Graduate Program, University of Texas at Austin, Austin, TX, USA. [2]Department of Molecular Biosciences, University of Texas at Austin, Austin, TX, USA. [3]Center for Systems and Synthetic Biology, University of Texas at Austin, Austin, TX, USA. [4]Livestrong Cancer Institutes, Dell Medical School, Austin, TX, USA. [5]Present address: Department of Molecular Biology and Genetics, Johns Hopkins University School of Medicine, Baltimore, MD, USA. ✉e-mail: dtaylor@utexas.edu; arlen@utexas.edu

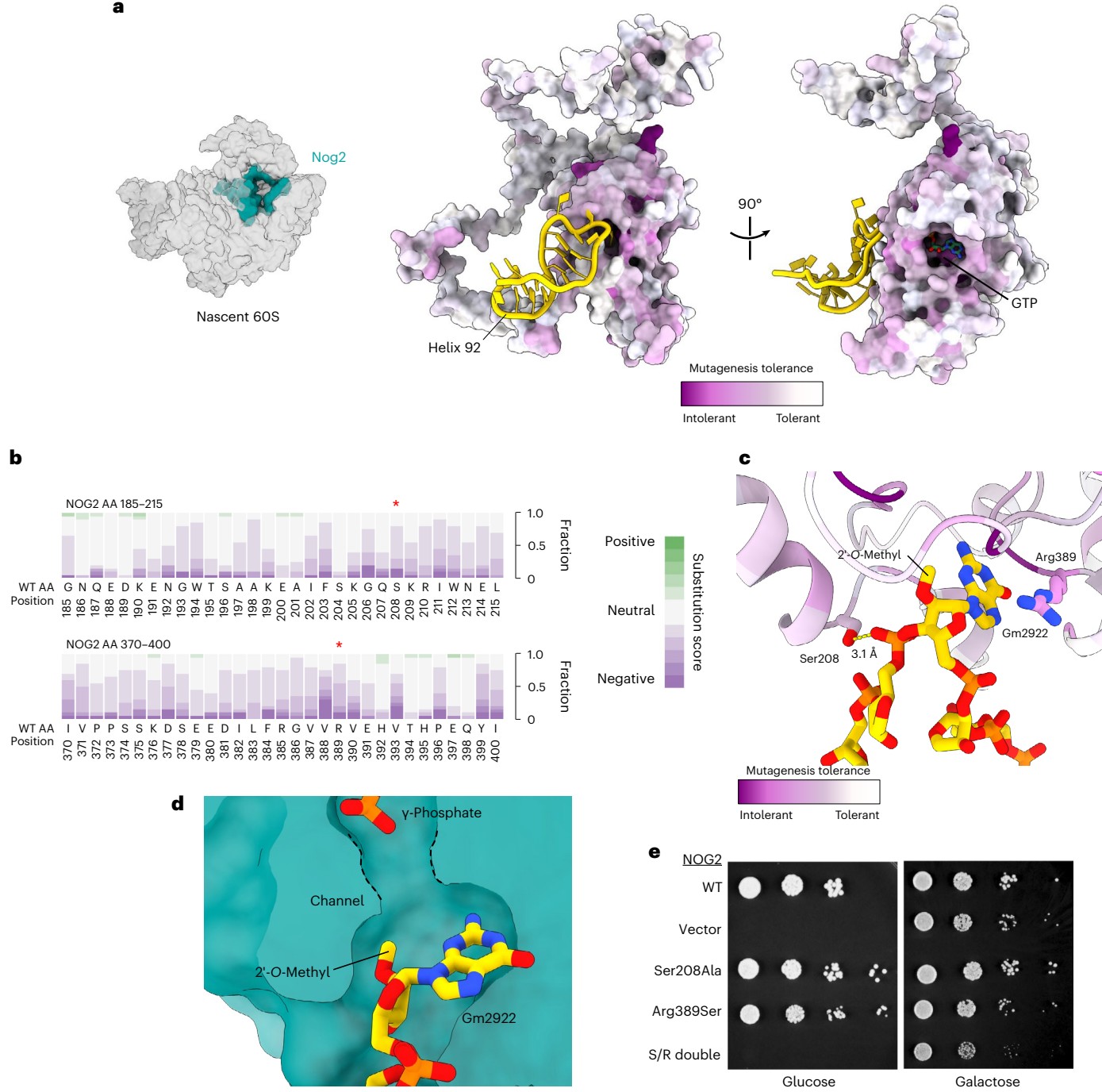

**Fig. 1 | Essential function of Nog2 depends on physical interaction with H92.**
**a**, Left, structure of Nog2 relative to pre-60S (accession no. PDB 3JCT) in 'crown view'. Right, Nog2 colored by per-residue average score of amino acid tolerance to mutation. H92 is colored gold. **b**, Cumulative per-residue fitness scores for all amino acid substitutions at H92-interacting regions of Nog2. Residues Ser208 and Arg389 are denoted by red asterisks. **c**, Cartoon representation of Nog2 residues interacting with H92, colored as in **a**. **d**, Cutaway surface representation of Nog2, with Gm2922 flipped into the active site channel. The γ-phosphate of GTP is also shown. **e**, Serial dilution assays to test complementation by the indicated *NOG2* alleles in a repressible *NOG2* strain (glucose, endogenous *NOG2* repressed; galactose, expressed).

## Results

### Nog2 function is connected to physical interaction with Gm2922

To identify functionally important regions of Nog2, we generated a library of ~7,800 *NOG2* codon variants (equal to ~50% of possible amino acid substitutions) via mutagenic PCR. We introduced this library into a yeast strain in which we could select for functional mutants, where the endogenous *NOG2* gene was under control of the glucose-repressible *GAL1* promoter. We sequenced and analyzed *NOG2* variants with a deep mutagenesis scanning pipeline[8] (Extended Data Fig. 1a–d) and mapped intolerance to mutation onto the structure of Nog2 (Fig. 1a). Surprisingly, we found a distinct resistance to mutation in the amino acid residues interacting with H92 (Fig. 1b). More strikingly, several mutation-intolerant residues surround Gm2922 (Fig. 1c). Published

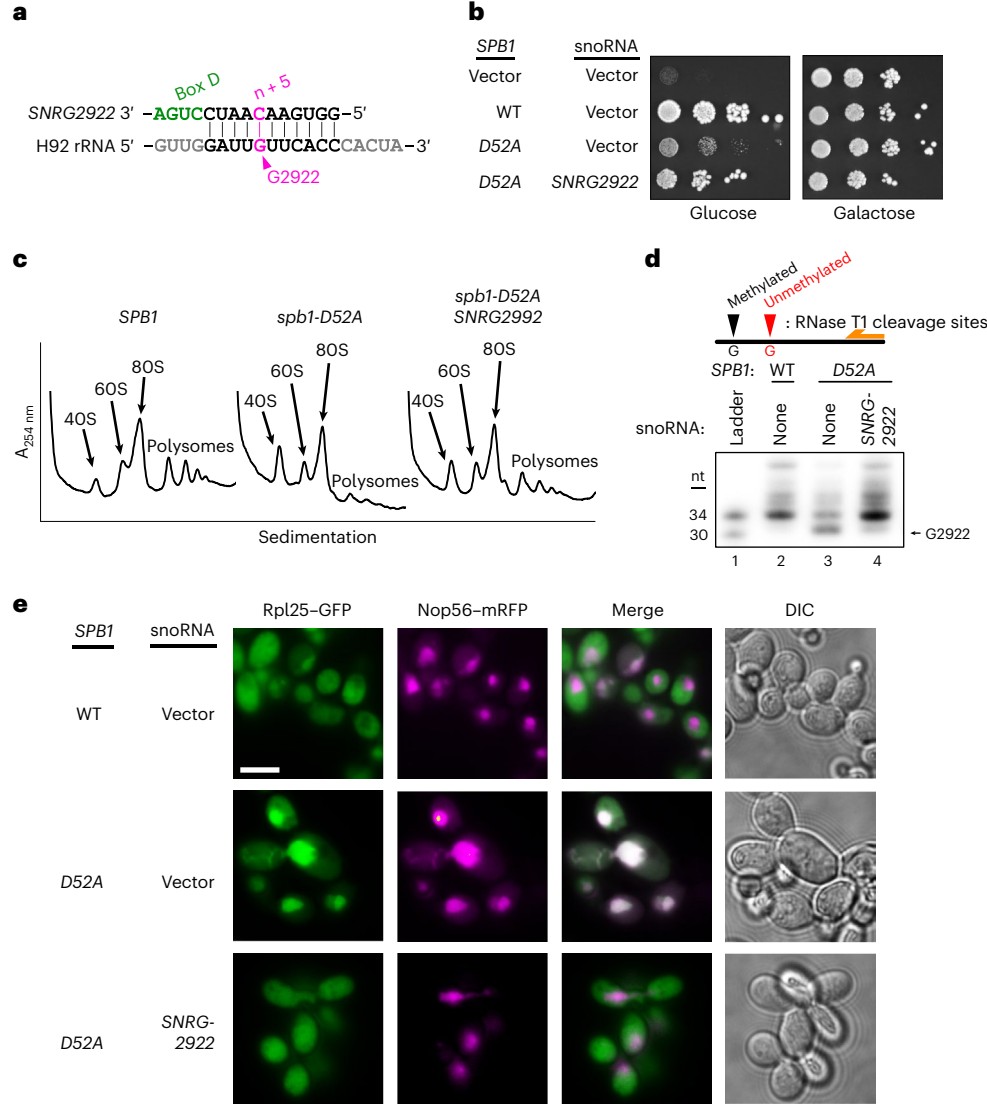

**Fig. 2 | 2′-O-Methylation of Gm2922 is necessary and sufficient for robust cell growth and nuclear export of the nascent large ribosomal subunit. a,** Design of synthetic snoRNA gene *SNRG2922* to direct 2′-O-methylation of Gm2922, catalyzed by the box C/D snoRNP complex at the fifth nucleotide upstream of box D, depicted in green. **b,** Expression of *SNRG2922* overcomes the growth defect of the methyltransferase-defective *spb1-D52A* mutant in a repressible *SPB1* strain (glucose, endogenous *SPB1* repressed; galactose, expressed). **c,** Sucrose gradient sedimentation profiles for the indicated *SPB1* and snoRNA alleles. **d,** Primer extension assay to probe 2′-O-methylation at G2922. Autoradiography of PAGE-separated primer extension products of RNAse T1-digested RNA. The experiment was repeated twice with similar results. **e,** Fluorescence microscopy to monitor localization of large subunit (Rpl25–enhanced (e)GFP) and nucleolus (Nop56–basic red fluorescent protein (mRFP)). The experiment was repeated twice with similar results. Scale bar, 5 μm. DIC, differential interference contrast.

structures of Nog2-bound, pre-60S intermediates show that Gm2922 is flipped from its canonical position[9] by a 2′-*endo* pucker, into a channel gating the active site of Nog2 (Fig. 1c,d and Extended Data Fig. 2a). Modeling the 2′-O-methyl of Gm2922, which is clearly resolved in a previous cryo-EM structure of Nog2 (ref. [10]), reveals how it is accommodated at the entrance to the active site (Fig. 1d and Extended Data Fig. 2b). In addition, Arg389 of Nog2, which resisted amino acid substitutions (Fig. 1b), stabilizes Gm2922 through a cation-π interaction with the guanosine base (Fig. 1c and Extended Data Fig. 2c)[11,12]. On the opposite side of the channel, Ser208 forms a hydrogen bond with the 5′-phosphate of Ψ2923 (Fig. 1c and Extended Data Fig. 2c). As expected, Ser208 was also resistant to mutation in our analysis (Fig. 1b). To directly assess the functional importance of Nog2 binding to Gm2922, we tested the ability of the strongly disfavored *NOG2* variants Ser208Ala and Arg389Ser to complement loss of endogenous *NOG2*. Although we observed no obvious growth

defect in the Ser208Ala variant and a modest growth reduction in the Arg389Ser variant, combining these mutations rendered a lethal variant (Fig. 1e). We confirmed that loss of function was not due to lack of Nog2 expression (Extended Data Fig. 1d). Taken together, these results demonstrate that at least two amino acids physically interacting with Gm2922 are interdependent and important for the essential function of Nog2.

## 2′-O-Methylation of G2922 is critical for ribosome assembly

G2922 is 2′-O-methylated by the conserved methyltransferase Spb1 (refs. [13,14]), which is essential in yeast[15]. Previous studies showed that a catalytically dead variant of Spb1 is detrimental to cell growth[13,14], but did not address the role of its target base. To specifically test the importance of methylating G2922, we engineered an artificial small nucleolar (sno)RNA to guide methylation of G2922 (Fig. 2a) via the box C/D snoRNP complex[16]. We introduced *SNRG2922* into a yeast

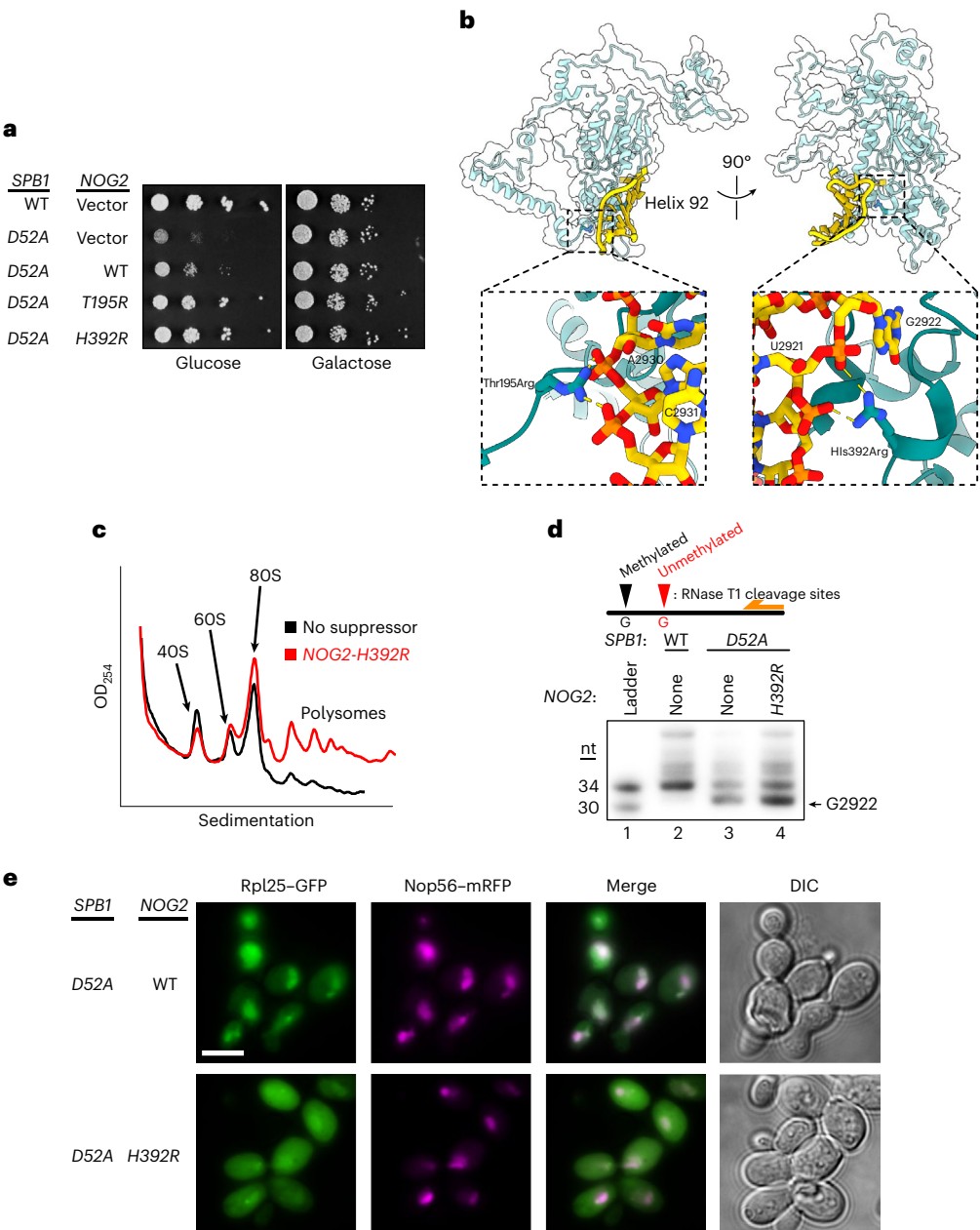

**Fig. 3 | Single amino acid changes in Nog2 restore robust growth and nuclear export in the absence of Gm2922. a**, Genetic suppression of a lack of Gm2922 by the indicated *NOG2* alleles in a repressible *SPB1* strain (glucose, endogenous *SPB1* repressed; galactose, expressed). **b**, Cartoon and surface representation of Nog2, with suppressing amino acid changes modeled (accession no. PDB 3JCT). Left inset, suppressing amino acid change Thr195Arg modeled to show new electrostatic interactions with the phosphates of the H92 bases A2930 and C2931. Right inset, suppressing amino acid change His392Arg modeled similarly to show new electrostatic interactions with the phosphates of the H92 bases

U2921 and G2922. **c**, Sucrose gradient sedimentation profile from *spb1-D52A* cells suppressed by *NOG2-H392R*, contrasted with empty vector (black trace same as shown in Fig. 2c). **d**, Primer extension assay to probe 2′-*O*-methylation at G2922 in WT cells, cells expressing methyltransferase-defective *spb1-D52A* and *spb1-D52A* suppressed by *NOG2-H392R*. Lanes 2 and 3 are the same samples as loaded in Fig. 2d. The experiment was repeated twice with similar results. **e**, Fluorescence microscopy to monitor location of large subunit (Rpl25–eGFP) and nucleolus (Nop56–mRFP) when *spb1-D52A* is suppressed by WT *NOG2* or *NOG2-H392R*. The experiment was repeated twice with similar results. Scale bar, 5 μm.

strain expressing a methyltransferase-deficient mutant of *SPB1*, *spb1-D52A*, which cannot methylate G2922 (refs. [13,14]). Strikingly, expression of *SNRG2922* supported near wild-type (WT) growth in the *spb1-D52A* strain (Fig. 2b). To verify that improved growth correlated with restoration of the ribosome assembly, we analyzed the polysome profiles of WT *SPB1*, *spb1-D52A* and the *SNRG2922*-complemented *spb1-D52A* strains. The strong 60S synthesis defect of the *spb1-D52A* mutant, indicated by low levels of free 60S and polysomes, was greatly alleviated by expressing

*SNRG2922* (Fig. 2c). To confirm that *SNRG2922* directs methylation of Gm2922, we designed a primer extension assay, taking advantage of the fact that 2′-*O*-methylated rRNA is resistant to cleavage by RNASE T1 (ref. [14]) (Extended Data Fig. 3a). G2922 was susceptible to cleavage in the *spb1-D52A* strain, but resistant to cleavage when *SNRG2922* was expressed, indicating that *SNRG2922* restores 2′-*O*-methylation of Gm2922 (Fig. 2d). Together, these results demonstrate that 2′-*O*-methylation of G2922 is critical for ribosome biogenesis in yeast.

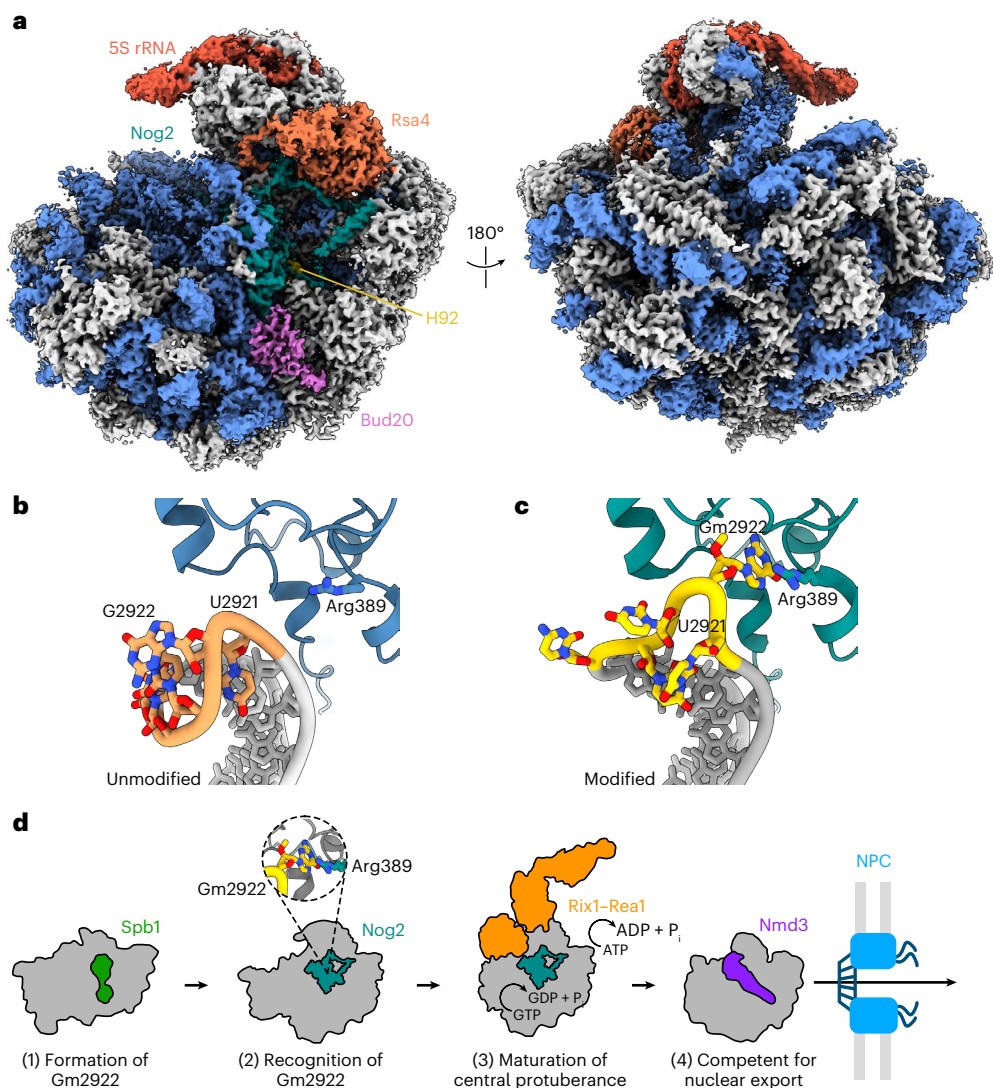

**Fig. 4 | Nog2 does not engage with unmodified G2922. a**, Cryo-EM structure of pre-60S in complex with Nog2 bound to H92 from methyltransferase-defective *spb1-D52A* mutant cells in which G2922 is unmodified (crown view, left). The unrotated 5S rRNA is indicated, along with Nog2, H92, Rsa4 and Bud20. **b,c**, Atomic models of unmodified G2922 and Nog2 (**b**) and modified G2922 and Nog2 (accession no. PDB 3JCT) (**c**). **d**, Cartoon model for Nog2 gating of 60S assembly and nuclear export, dependent on methylation of G2922. NPC, nuclear pore complex.

## Nog2 monitors the methylation status of G2922

We hypothesized that Nog2 directly monitors the methylation of G2922. As Nog2 is required for export of the pre-60S[15,17], we reasoned that failure to methylate G2922 would interrupt nuclear export. To test this hypothesis, we monitored the intracellular localization of Rpl25–green fluorescent protein (GFP), an established reporter for nuclear export of the pre-60S[18]. Although Rpl25 was predominantly cytoplasmic in cells expressing WT *SPB1*, reflecting the localization of mature ribosomes, the methyltransferase-deficient *spb1-D52A* induced nucleolar localization of Rpl25–GFP, suggesting that lack of 2′-*O*-methylation leads to nucleolar retention of the pre-60S (Fig. 2e). Remarkably, expression of *SNRG2922* restored localization of Rpl25–GFP to the cytoplasm (Fig. 2e), indicating that the single 2′-*O*-methylation of Gm2922 is required for efficient nuclear export of the pre-60S.

We further predicted that if Nog2 associates weakly with an unmodified G2922, overexpression of Nog2 would overcome cellular dependence on Gm2922. Indeed, overexpression of *NOG2* weakly suppressed the growth defect of methyltransferase-deficient *spb1-D52A* cells (Fig. 3a). Similarly, we reasoned that variants of Nog2 capable of binding more strongly to H92 would more efficiently bypass the growth defect caused

by lack of G2922 methylation. To directly test this idea, we designed a *NOG2* variant, Thr195Arg, a positively charged substitution predicted to bind more strongly to the negatively charged H92 phosphodiester backbone (Fig. 3b, left). Strikingly, *NOG2-T195R* restored growth in the *spb1-D52A* strain to near WT levels (Fig. 3a). In parallel, we screened our *NOG2* variant library for randomly generated suppressors of *spb1-D52A* and identified a second *NOG2* variant, His392Arg, which also restored growth to near WT levels (Fig. 3a). This variant is also predicted to increase electrostatic binding with the phosphodiester backbone of H92 (Fig. 3b, right). We confirmed that expression of *NOG2-H392R* restored 60S biogenesis in the *spb1-D52A* strain (Fig. 3c), despite the absence of 2′-*O*-methylated G2922 (Fig. 3d). Moreover, expression of *NOG2-H392R* restored nuclear export of the pre-60S (Fig. 3e). Our finding that single amino acid changes in Nog2 can bypass cellular dependence on Gm2922 strongly suggests that Nog2 directly monitors formation of Gm2922 to gate nuclear export of the large ribosomal subunit.

## Absence of 2′-*O*-methylation traps a nucleolar pre-60S

To investigate the molecular defects caused by lack of G2922 modification, we genetically tagged Nog2 and immunopurified, Nog2-bound

**Table 1 | Cryo-EM data collection, refinement and validation statistics**

| | Nog2-3×FLAG unmodified G2922 (EMD-26485; PDB 7UG6) |
|---|---|
| **Data collection and processing** | |
| Magnification | ×29,000 |
| Voltage (kV) | 300 |
| Electron exposure (e⁻ Å⁻²) | 70 |
| Defocus range (µm) | −1.5 to −2.5 |
| Pixel size (Å) | 0.81 |
| Symmetry imposed | C1 (that is, none) |
| Initial particle images (no.) | 403,566 |
| Final particle images (no.) | 15,954 |
| Map resolution (Å) | 2.9 |
| FSC threshold | 0.143 |
| Map resolution range (Å) | 2–10 |
| **Refinement** | |
| Initial model used (PDB accession no.) | 3JCT |
| Model resolution (Å) | 2.9 |
| FSC threshold | 0.5 |
| Model resolution range (Å) | Not applicable |
| Map sharpening $B$ factor (Å²) | 19.0 |
| Model composition | |
| Nonhydrogen atoms | 140,993 |
| Protein residues | 8,739 |
| Nucleotides | 3,338 |
| Ligands | Mg²⁺: 1 |
| $B$ factors (Å²) | |
| Protein | 38.9 |
| Ligand | 62.99 |
| R.m.s. deviations | |
| Bond lengths (Å) | 0.010 |
| Bond angles (°) | 0.880 |
| Validation | |
| MolProbity score | 1.88 |
| Clashscore | 11.12 |
| Poor rotamers (%) | 0.01 |
| Ramachandran plot | |
| Favored (%) | 95.37 |
| Allowed (%) | 4.62 |
| Disallowed (%) | 0.01 |

pre-60S subunits from cells expressing methyltransferase-deficient *spb1-D52A*, or WT *SPB1* as a control (Extended Data Fig. 4a). Semiquantitative tandem mass spectrometry (MS–MS) analysis showed that Nog2 particles from *spb1-D52A* cells are depleted of components of the Rix1–Rea1 complex required for maturation of the 60S central protuberance[19], as well as of Sda1, a ribosome assembly factor involved in recruitment of the Rix1–Rea1 complex[12], and nuclear export of the pre-60S subunit[20] (Extended Data Fig. 4b). Furthermore, we observed that unmodified subunits are deficient in Arx1 and Bud20, nonessential but important assembly factors that bind the nuclear subunit

and contribute to its efficient cytoplasmic export[21–23] (Extended Data Fig. 4a,b). In turn, Nog2 particles from *spb1-D52A* cells are enriched for nucleolar factors, including Spb1, consistent with arrest of an earlier pre-ribosomal complex (Extended Data Fig. 4b). Failure to methylate G2922 therefore traps a nucleolar intermediate, supporting our conclusion that Gm2922 is important for the pre-60S to transition from the nucleolus to the nucleoplasm.

**Unmodified G2922 does not flip into active channel of Nog2**

Our results strongly suggest that Nog2 binding to H92 depends on methylation of G2922. To directly visualize the consequences of unmethylated G2922, we used cryo-EM to solve the structure of an unmodified pre-60S subunit bound by Nog2 at an overall resolution of 2.9 Å (Fig. 4a, Extended Data Figs. 5 and 6 and Table 1). In our structure, density for Nog2 largely agrees with prior structure determinations[10,12,19]. However, switch I and switch II (motifs G2 and G3) are apparently ordered around GDP in complex with a magnesium ion (Extended Data Fig. 7a), suggesting that Nog2 has hydrolyzed GTP, as also reported in recent unpublished work[24]. Most strikingly, the unmodified G2922 is not flipped into the active site channel of Nog2, unlike the modified Gm2922 in previously solved structures (Fig. 4b,c). Instead, when G2922 lacks a 2′-$O$-methyl group, H92 closely matches the conformation it adopts within the mature large ribosomal subunit, where Gm2922 stacks on Ψ2923 (Extended Data Fig. 7b)[9]. This result unambiguously shows that methylation of G2922 is critical for its engagement with the active site channel of Nog2 and suggests that Nog2-catalyzed GTP hydrolysis is triggered in the absence of Gm2922. Thus, Nog2 directly assesses the methylation status of Gm2922, gating ribosome biogenesis before nuclear export of the nascent 60S subunit.

## Discussion

Our work demonstrates how a single nucleotide modification serves as a structural checkpoint in ribosome assembly. We have shown that recognition of Gm2922 by the essential GTPase Nog2 couples structural maturation of the ribosome to Nog2-dependent nuclear export. Initially, methylation of G2922 by Spb1 prepares the RNA of the proto A-site for binding by Nog2. Nog2 then assesses the modification status of Gm2922 to ensure that only ribosomes with a methylated G2922 progress through the biogenesis pathway. Stable binding of Nog2 to H92 is a prerequisite for recruitment of the Rix1–Rea1 complex, which drives maturation of the central protuberance. Subsequent GTP hydrolysis by Nog2 promotes its release from the pre-60S[25]. Since Nog2 occludes the binding site for the export adapter protein Nmd3, once Nog2 is released, Nmd3 is free to bind to escort the pre-60S to the cytoplasm (Fig. 4e).

In human cells, G4469 (the equivalent of G2922) is also 2′-$O$-methylated[26] and, since Nog2 (GNL2 in humans) is highly conserved between yeast and humans, we anticipate a similar functional and structural role for Gm4469 in assembly of the human large ribosomal subunit. Our work clearly demonstrates the importance of G2922 modification for ribosome assembly. However, as the A-loop plays a critical role in accommodating amino-acyl tRNAs during translation elongation and chemical modification of the A-loop is found in ribosomes from all extant organisms, it is tempting to speculate that modification of Gm2922 or neighboring nucleotides is also important for translation. For example, rRNA modifications of the A-site loop could alter the rate or accuracy of the translating ribosome. Surprisingly, we found that *NOG2* mutants that bypass the requirement for G2922 methylation appear to have minimal impact on cell growth. Nevertheless, it remains possible that modification of the A-loop has a greater impact on translation under suboptimal growth conditions. Future work will address the biochemical consequences of lacking A-site modifications, as well as the environmental conditions under which they are most critical for survival or robust growth.

Surveillance of rRNA modification by orthologous GTPases may well extend beyond nuclear ribosome assembly in eukaryotes.

Several studies examining the human mitochondrial ortholog of Nog2, MTG1 (GTPBP7), suggest that this enzyme binds to the maturing H92 of the mitochondrial large subunit and may directly sense modifications of universally conserved A-loop bases including Um3039 and Gm3040 (refs. [27–29]). Also, the Nog2 bacterial ortholog RbgA interacts closely with H92 in *Bacillus subtilis* ribosome assembly[30], where the A-loop base Gm2582 (analogous to *Saccharomyces cerevisiae* Gm2922) is known to be 2′-*O*-methylated[31,32]. The structures of bacterial and mitochondrial Nog2 orthologs have conserved histidine residues thought to play a role in catalysis[33], either by coordinating a water molecule for the first nucleophilic attack step of GTP hydrolysis or controlling access to the γ-phosphate (Extended Data Fig. 7c). Intriguingly, Gm2922 fills an analogous position in the active site channel of Nog2, where it potentially occludes water molecule access to the γ-phosphate of GTP. This hypothesis is supported by our observation that GDP is present in the active site of Nog2 when Gm2922 is absent, although future biochemical experiments are required to fully validate this speculation. Although these enzymes have experienced more than one billion years of evolutionary divergence, their common structural features and binding location on the nascent large ribosomal subunit raise the exciting possibility that RNA modifications play a critical role in regulating their enzymatic functions. Taken together, our results suggest a new paradigm for rRNA modifications in gating assembly of this important macromolecular machine.

## Online content

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

## Methods

### Strains, plasmids and cell growth

Yeast strains, plasmids and oligonucleotides used in the present study are listed in Supplementary Data 1. All yeast strains are derivatives of BY4741 and were grown at 30 °C. Golden Gate assemblies were performed using BsaI-HF or Esp3I enzymes from New England Biolabs. All site-directed mutagenesis was performed using inverse PCR or Gibson assembly with mutagenic oligonucleotides. All plasmid assemblies requiring a PCR-amplified insert were verified via Sanger sequencing. Yeast transformations were all conducted via the poly(ethylene glycol) lithium acetate method[34]. Yeast Toolkit (YTK) plasmids were a gift from J. Dueber (Addgene, kit no. 1000000061)[35].

Strain AJY4406 (*NatR::pGAL1:3xHA-NOG2*) was generated via PCR amplification of a modular *NatR:pGAL1:3xHA* cassette with oligonucleotides containing *NOG2* promoter and coding sequence, as previously described[36]. Strain AJY4425 (*KanMX::pGAL1:SBP1*) was constructed similarly. Strain AJY4427 (*KanMX::pGAL1:SPB1 NOG2Δintron-3xFLAG-6xHis*) was generated via a YTK-compatible Cas9 protocol described previously[37]. A Cas9 small guide (sg)RNA (5′-TTTGATTCTTTCCAAC CAAG-3′) was designed to target the intron within *NOG2* and assembled into sgRNA dropout vector pAJ4247 to construct the constitutive Cas9-sgRNA plasmid pAJ4810; strain AJY4425 was then co-transformed with pAJ4810 and a linearized repair template corresponding to *NOG2* without an intron and containing a carboxy-terminal 3xFLAG-6xHis tag (derived from digestion of pAJ4853 with EagI). Candidate *NOG2 Δintron-3xFLAG* recombinants were screened for loss of the *NOG2* intron by colony PCR and for Nog2-3xFLAG expression by western blotting. A successful recombinant was finally streaked on yeast peptone galactose medium and colonies screened for *URA3* auxotrophy to verify loss of pAJ4810.

### PCR mutagenesis of *NOG2*

Plasmid pAJ4395 was constructed by amplifying the *NOG2* locus (~500 nt upstream of the *NOG2* start codon to ~80 nt downstream of the *NOG2* stop codon) from genomic DNA purified from yeast strain BY4741 and cloning it into vector pAJ5103 (*CONS-sfGFP dropout-CON1 LEU2 CEN AmpR*) assembled from YTK parts[35]. The internal BsaI site in the *NOG2* coding sequence was replaced with a synonymous coding sequence (G444>A), and the single intron within the *NOG2* locus was removed. The *NOG2* open reading frame was mutagenized using Taq polymerase (New England Biolabs), with amplifying primers AJO3460 (5′-TAGCGGTCTCAACTAAATATCAGGGGAGGAATAATACA-3′) and AJO3461 (5′-TAGCGGTCTCACGGTTGTCCGTTTTTACCTCTA-3′), containing BsaI sites for Golden Gate assembly. To alleviate bias in the type and rate of mutations, dNTP concentrations were adjusted to 0.2 mM dATP and dGTP and 1 mM dTTP and dCTP[38]. After five amplification cycles, manganese chloride was added to a final concentration of 500 μM[39]. PCR was then continued for another 20 cycles. The promoter (~500 nt) and terminator (~200 nt) regions of *NOG2* were amplified separately from BY4741 genomic DNA, with a high-fidelity polymerase, using oligonucleotides AJO3454 (5′-TAGCGGTCTCAAACGCGGAAAAGAAGGAGAGTCTG-3′) × AJO3459 (5′-TAGCGGTCTCATAGTGGTTTTGCTGCAAGTC-3′) (*NOG2* promoter) and AJO3458 (5′-TAGCCGTCTCATCGGTCTCAACCGGTGTATAATAT TAATTTAAGATTACA-3′) × AJO3455 (5′-TAGCGGTCTCACAGCTCTAGAGG ATCTAACAATGAACTTAAATCAACAACACG-3′) (*NOG2* terminator), all of which contain compatible 5′-BsaI sites for directional assembly.

Mutagenized *NOG2* was combined with the PCR products encoding its promoter and terminator regions in a BsaI Golden Gate assembly reaction and with the YTK ConS/1 GFP dropout vector pAJ5103. The assembled product was used to transform competent DH5α *Escherichia coli* (New England Biolabs) and transformants (~10⁵) were selected on ampicillin plates. Fewer than 1 in 10⁴ colonies expressed GFP, indicating successful library assembly. Several colonies were sequenced to verify the presence, quantity and uniformity

of mutations, and the rest were pooled and bulk DNA prepared via Midiprep (QIAGEN).

The mutagenized library prep was digested with BsaI to eliminate any remaining vector background and used to transform strain AJY4406. Transformants (~5 × 10⁴) were selected on synthetic solid medium containing galactose and lacking leucine, pooled and frozen in a glycerol stock. Cells from this stock were plated on to synthetic medium containing glucose and lacking leucine, thereby selecting transformants that could complement repression of the endogenous *NOG2* locus. Colonies viable on glucose (~5 × 10⁴) were pooled and frozen in a glycerol stock.

To estimate sequencer errors, a WT control was carried out in which strain AJY4406 was transformed with plasmid pAJ4395, harboring the unmutagenized *NOG2* coding sequence. These transformants were grown in synthetic liquid medium lacking leucine, with galactose or glucose as a carbon source for the input and selection controls, respectively.

### Sequencing and analysis of mutagenized yeast libraries

Plasmid extraction and all subsequent steps were performed in duplicate. Briefly, plasmid DNA was extracted from ~3 × 10⁹ cells (Zymoprep Yeast Plasmid Miniprep, Zymo Research) and amplified with vector-specific oligonucleotides AJO3514 (5′-TAACTGCCTTGA TCTGTCGG-3′) and AJO3515 (5′-TGGTAGAGCCACAAACAGC-3′). These amplicons were subsequently PCR amplified using 15 interleaved primer pairs (Supplementary Data 1), each yielding an amplicon of 150 bp, flanked by sequence for Illumina adapter priming. Amplicons for sequencing were pooled according to sample and multiplexing NEBNext Multiplex Illumina adapters (New England Biolabs) were attached in a final PCR step. The pools were sequenced in a single run on an Illumina NextSeq platform, using 150-bp paired-end reads. Reads were aligned to the *NOG2* open reading frame using BowTie2 (ref. [40]) and split into their corresponding tiles with a customized Python script. Mutations were counted and normalized using the TileSeq Analysis package (v.1.5)[8]. These mutational counts were analyzed and the complete phenotypic profile imputed and displayed using the POPCode Analysis Pipeline[8]. Customized Python scripts were used to calculate median fitness scores from the complete analysis and these scores were mapped on to the atomic model of Nog2. Evolutionary conservation scores were obtained from a ConSurf[41] prediction based on 495 orthologs of NOG2 obtained from the eggNOG database, v.4.5 (ref. [42]).

### Western blotting

Strain AJY4406 was transformed separately with plasmids pAJ5125, pAJ4853, pAJ5402, pAJ5403 and pAJ5404. Transformants were grown to stationary phase, diluted 1:20 and grown for 5 h at 30 °C. Cells were harvested and proteins extracted by incubating in 100 mM NaOH for 20 min on ice, then boiling in sodium dodecylsulfate (SDS)–polyacrylamide gel electrophoresis (PAGE) loading buffer for 5 min. Protein extracts were resolved on a 6–18% polyacrylamide gel, transferred to a nitrocellulose membrane and incubated overnight at 4 °C in tris-buffered saline (TBS) supplemented with 1% casein, containing primary antibodies rabbit anti-glucose-6-phosphate dehydrogenase (diluted 1:20,000) and rat anti-FLAG (Agilent, diluted 1:20,000). After two washes with TBS + Tween 20 (TBS-T) and one with TBS, the membrane was incubated for 30 min in TBS containing secondary antibodies goat anti-rabbit 680RD and goat anti-rat 800CW (LiCOR, each diluted 1:20,000). After three washes with TBS, secondary antibodies were visualized on a LiCOR Odyssey infrared scanner.

### Sucrose gradient sedimentation

Strain AJY4425 was co-transformed with the following paired combinations of plasmids: pAJ4841/pAJ5136, pAJ4842/pAJ5136, pAJ4842/ pAJ4852 and pAJ4842/pAJ4881. The transformants were inoculated into synthetic medium lacking leucine and histidine and containing

galactose and grown overnight at 30 °C to saturation. The transformants were then diluted 1:200 into synthetic medium lacking leucine and histidine and grown overnight at 30 °C to repress the expression of genomic *SPB1*. Cells from these cultures were finally diluted into the same synthetic medium, grown to mid-log phase (optical density at 600 nm ($OD_{600}$) ≈ 0.6,) then treated with 100 µg ml$^{-1}$ of cycloheximide and shaken for 30 min at 30 °C. Treated cells were centrifuged and stored at −80 °C. Thawed cells were washed and resuspended in lysis buffer (20 mM Hepes-KCl, pH 7.4, 100 mM KCl, 2 mM MgCl$_2$, 5 mM 2-mercaptoethanol, 100 µg ml$^{-1}$ of cycloheximide, 1 mM each of phenylmethylsulfonyl fluoride and benzamidine, and 1 µM each of leupeptin and pepstatin). Extracts were prepared by glass bead lysis (two 30-s rounds of vortexing) and clarified by centrifuging for 15 min at 18,000*g* and 4 °C. Then, 4.5 $OD_{260}$ units of clarified extract were loaded onto a 7–47% sucrose gradient made in the same buffer lacking the protease inhibitors. Gradients were centrifuged for 2.5 h at 202,000*g* in a Beckman SW40 rotor and the RNA content monitored at 254 nm using an ISCO Model 640 fractionator.

### RNAse T1 digestion and primer extension
Strain AJY4425 was co-transformed with the following paired combinations of plasmids: pAJ4841/pAJ5136, pAJ4842/pAJ5136, pAJ4842/pAJ4852 and pAJ4842/pAJ4881. The transformants were inoculated into synthetic medium lacking leucine and histidine and containing galactose and grown overnight at 30 °C to saturation. To repress the expression of genomic *PGAL1-SPB1*, the cells were diluted 1:200 into pre-warmed medium containing glucose and grown overnight at 30 °C to saturation. Finally, to dilute mature ribosomes synthesized before Spb1 depletion, the cells were again diluted 1:2,000 into fresh, pre-warmed SD-Leu/His medium containing glucose and cultured at 30 °C until the early exponential phase ($OD_{600}$ ≈ 0.3). Cells were harvested and stored at −80 °C. Total RNA was prepared using the acid–phenol–chloroform method as previously described[43]. To assay the methylation status of G2922, equivalent amounts of 25S rRNA were dried by speed vacuuming and resuspended in annealing buffer (40 mM 1,4-piperazinediethanesulfonic acid, pH 7.0, 400 mM NaCl and 1 mM EDTA). A mixture of RNA and $^{32}$P-labeled AJO3925 (5′-CCCAGCTCACGTTCCC-3′) was heated at 95 °C for 3 min and cooled to 4 °C using a ramp-down rate of 10 s per °C to promote annealing in a final volume of 25 µl. After annealing, 2 units of RNase T1 (Ambion) were added to 10 µl of the annealing reaction and incubated at 25 °C for 60 min. Reverse transcription reactions using MMLV reverse transcriptase (Invitrogen) were performed according to the manufacturer's instructions, using 4 µl of each RNase T1 digest reaction as a template in a final reaction volume of 10 µl. Reactions were carried out at 37 °C for 50 min followed by inactivation at 70 °C for 15 min. One volume (10 µl) of 2× Tris-borate–EDTA (TBE)–urea sample buffer (89 mM Tris, 89 mM boric acid, 2 mM EDTA, pH 8.0, 7 M urea, 12% Ficoll, 0.01% Bromophenol Blue and 0.02% xylene cyanol FF) was added to each sample and heated at 70 °C for 15 min. Then, 2 µl of each sample was separated on a 12% TBE–urea gel, dried on filter paper and exposed to a phosphoscreen overnight. Signal was detected by phosphorimaging on a GE Typhoon FLA9500.

### Fluorescence microscopy
Strain AJY4425 harboring pAJ4890 was co-transformed with the following paired combinations of plasmids: pAJ4841/pAJ5136, pAJ4842/pAJ5136, pAJ4842/pAJ4852, pAJ4842/pAJ4872 and pAJ4842/pAJ4881. Strains were grown to saturation in synthetic medium containing galactose and lacking uracil, leucine and histidine, then subcultured for 16 h in medium containing glucose to deplete endogenous Spb1. Cells were then freshly diluted 1:20 and grown for 4 h in medium containing glucose. Cells were fixed for 30 min with 3.7% freshly prepared formalin, washed 3× with cold 100 mM potassium phosphate, pH 6.4 and incubated for 5 min at room temperature with 0.1% Triton X-100.

DAPI was added to 2 µg ml$^{-1}$ and cells were incubated for another 3 min at room temperature, then washed 3× with cold phosphate-buffered saline, pH 7.4 and stored at 4 °C. Fluorescence and brightfield micrographs were recorded on a Nikon E800 microscope fitted with a ×100 Plan Apo objective and a Photometrics CoolSNAP ES camera controlled by NIS-Elements software.

### Immunopurification
Strain AJY4427 (*KanMX::pGAL1:SPB1 NOG2-3xFLAG-6xHis*) was transformed with WT SPB1 (pAJ4841) or *spb1-D52A* (pAJ4842) and grown for 24 h in medium lacking histidine and containing galactose, followed by 24 h of growth in medium containing glucose to deplete endogenous Spb1. Cells were then diluted in the same medium and grown to mid-log(phase) ($OD_{600}$ of 0.6). For analysis by PAGE and MS, cells were harvested and frozen at −80 °C. All subsequent steps were performed at 0–4 °C. Cell pellets were thawed and resuspended in immunoprecipitation (IP) binding buffer (50 mM Tris, pH 7.6, 100 mM NaCl, 5 mM MgCl$_2$, 0.05% NP-40 and 0.5 mM tris(2-carboxyethyl)phosphine hydrochloride (TCEP)) and broken with glass beads. The cell lysate was clarified by two subsequent centrifugation steps of 5 min at 3,000*g* followed by 20 min at 18,000*g*, and the lysate was incubated with αFLAG-conjugated agarose beads (Millipore-Sigma) for 2 h with rotation at 4 °C. The lysate was removed and the beads were washed 3× with IP wash buffer (50 mM Tris pH 7.6, 100 mM NaCl, 5 mM MgCl$_2$, 0.01% octyl-β-glucoside and 0.5 mM TCEP), after which the Nog2-3xFLAG complex was eluted for 90 min with IP wash buffer containing 350 µg ml$^{-1}$ of 3×FLAG peptide. For analysis by cryo-EM, cells were flash frozen as a 75% cell slurry in IP buffer and broken under cryogenic conditions in a Retsch mixer mill with 6 cycles of 3 min at 15 Hz. Subsequent steps were as described for analysis by PAGE and MS above. Eluates were either frozen and stored for PAGE and MS analysis or else directly applied to cryo-EM grids.

### Mass spectrometry
Protein identification was provided by the University of Texas at Austin Center for Biomedical Research Support Proteomics Facility (RRID:SCR_021728). Samples were excised from SDS–PAGE gels, trypsin digested, desalted and run on a Dionex LC and Orbitrap Fusion 2 for liquid chromatography–tandem MS for 30 min (single band) or 120 min (whole lane) and analyzed using PD 2.2 and Scaffold 5. For semiquantitative analysis, peptide spectral counts were normalized to protein molecular mass and *P* values for enrichment were calculated from a paired, two-sided Student's *t*-test using the Cyber-T web server[44] without Bayesian correction.

### Cryo-EM
Quantifoil R1.2/1.3 grids coated with an ultrathin layer of amorphous carbon (Electron Microscopy Sciences) were glow discharged for 1 min at 25 mA. Using a Mark IV VitroBot (FEI), a sample was applied to freshly glow-discharged grids at 4 °C and 100% humidity, immediately blotted for 2 s at a force setting of 0, then plunged into liquid ethane and stored under liquid nitrogen until data collection. Microscopy data were collected in two separate rounds at the University of Texas at Austin Sauer Structural Biology Center on a Titan Krios microscope (FEI) operating at 300 kV, equipped with a K3 Summit direct electron detector (Gatan). To alleviate particle orientation bias observed in the first collection, 6,768 videos were recorded at 0° tilt and 6,732 videos at 30° tilt. In the second round of data collection, 12,456 videos were collected at 0° tilt, for a total of 25,956 videos. Videos were acquired using SerialEM v.3.8 (ref. [45]) and recorded as 20 frames over 4 s for a total electron dosage of ~70 e$^-$ Å$^{-2}$. Videos were recorded at a nominal magnification of ×22,500, a pixel size of 0.81 Å and over a defocus range of −1.5 µm to −2.5 µm. Using cryoSPARC Live for on-the-fly processing, videos were motion corrected and dose weighted, contrast transfer function estimates performed and a total of 293,350 particles picked using a 60S template. Following on-the-fly two-dimensional (2D) classification to separate clean

pre-60S projections from unassignable or 'junk' classes, particles were exported to cryoSPARC[46] (v.3.2), where multiple subsequent rounds of 2D classification resulted in a total of 194,497 particles. These particles were used for the initial reconstruction and three-dimensional (3D) heterogeneous refinement to separate a total of 120,722 nucle(ol)ar particles with at least partial Nog2 occupancy, which were subjected to consensus nonuniform refinement. Using the csparc2star.py function in pyEM[47], these particles were exported to RELION[48] v.3.1.3, where a 3D classification scheme using a soft mask around Nog2/H92 was used to further separate 86,273 Nog2-bound particles. These particles were re-imported into cryoSPARC v.3.2 for a 3D classification scheme with a soft mask around Rsa4, which separated 15,954 particles with Nog2 at high resolution. Separation of Rsa4-bound Nog2 particles was key to obtaining the highest-possible resolution map of both Nog2 and the 5S particle, which is flexible before Rea1-mediated rotation. To further improve the map, four separate masks were generated covering the 5S particle and three other sections of the complex ('Core & L1-stalk', 'Foot' and 'P-stalk') and used for particle subtraction combined with local refinement in cryoSPARC v.3.2 (Extended Data Fig. 5). The resulting local maps were combined in ChimeraX[49]. Fourier shell correlation (FSC) calculations were also carried out in cryoSPARC v.3.2 after final refinement (Extended Data Fig. 6).

### Modeling, refinement and graphics

The Protein Data Bank (PDB) accession no. PDB 3JCT (ref. [10]) was first rigid-body docked into the final, unsharpened map, using UCSF ChimeraX v.1.0 (ref. [49]). Chains for uL11 and Alb1 were added from accession no. PDB 6YLH (ref. [12]). The rRNA of H92 was fit by hand into the corresponding map density using COOT v.1.0 (ref. [50]). The model was relaxed using flexible molecular dynamics fitting with ISOLDE v.1.4 (ref. [51]), then finally subjected to real-space refinement as implemented in PHENIX v.1.19.1 (ref. [52]). All molecular graphics were prepared with UCSF ChimeraX v.1.0 (ref. [49]).

### Reporting summary

Further information on research design is available in the Nature Portfolio Reporting Summary linked to this article.

### Data availability

The raw sequencing data are deposited in the Sequencing Read Archive with BioProject accession no. PRJNA892951. The structure of the Nog2-3×FLAG unmodified pre-60S subunit and its associated atomic coordinates have been deposited into the Electron Microscopy Data Bank (EMDB) and the PDB as accession nos. EMD-26485 and PDB 7UG6, respectively. All other data are available in the manuscript or supplementary materials. Requests for strains or plasmids will be fulfilled by the lead contact author, A.W.J., upon request. Source data are provided with this paper.

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

### Acknowledgements

We thank A. Brillot for assistance with cryo-EM and data acquisition. We also thank I. Hoskins for assistance with designing primers for TileSeq, J. Weile for helpful discussion and J. Erzberger and colleagues for sharing their unpublished work with us. In addition, we thank B. Xhemalce and J. Huibregtse for helpful comments on the manuscript and all members of A.W.J.'s, D.W.T.'s and Sarinay-Cenik labs for helpful discussions. This work was supported in part by the National Institute of General Medical Sciences of the National Institutes of Health (grant nos. R35GM237237 to A.W.J. and R35GM138348 to D.W.T.), Welch Foundation Research Grant to D.W.T (no. F-1938), and a Robert J. Kleberg, Jr. and Helen C. Kleberg Foundation Medical Research Grant to D.W.T. D.W.T is a CPRIT Scholar and an American Cancer Society Research Scholar supported by the Cancer Prevention Research Institute of Texas (no. RR160088) and the American Cancer Society (no. RSG-21-050-01-DMC).

### Author contributions

J.N.Y. and A.W.J. conceived the project. J.N.Y., J.P.K.B. and J.J.B. performed the experiments and visualized the results. J.N.Y., D.W.T. and A.W.J. wrote the original draft of the manuscript. J.N.Y., J.P.K.B., J.J.B., D.W.T. and A.W.J. edited and reviewed the final manuscript. D.W.T. and A.W.J. supervised and secured funding for the research.

### Competing interests

The authors declare no competing interests.

## Additional information

**Extended data** is available for this paper at https://doi.org/10.1038/s41594-022-00891-8.

**Correspondence and requests for materials** should be addressed to David W. Taylor or Arlen W. Johnson.

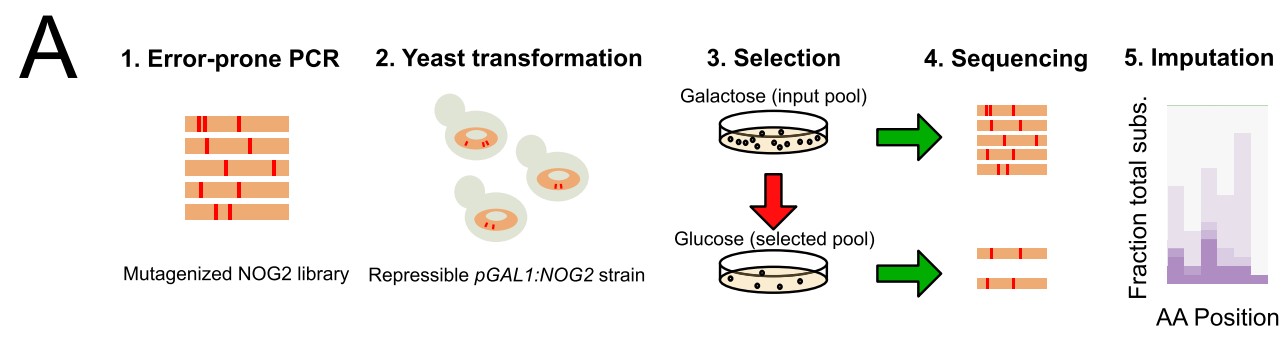

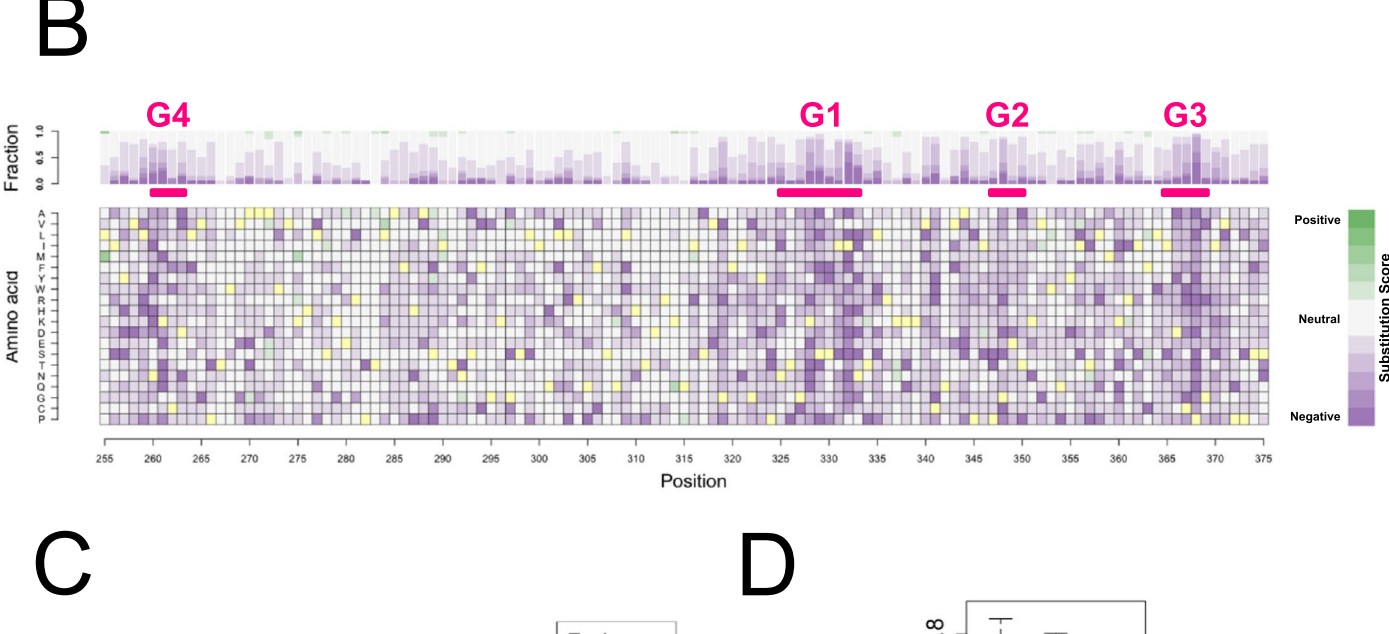

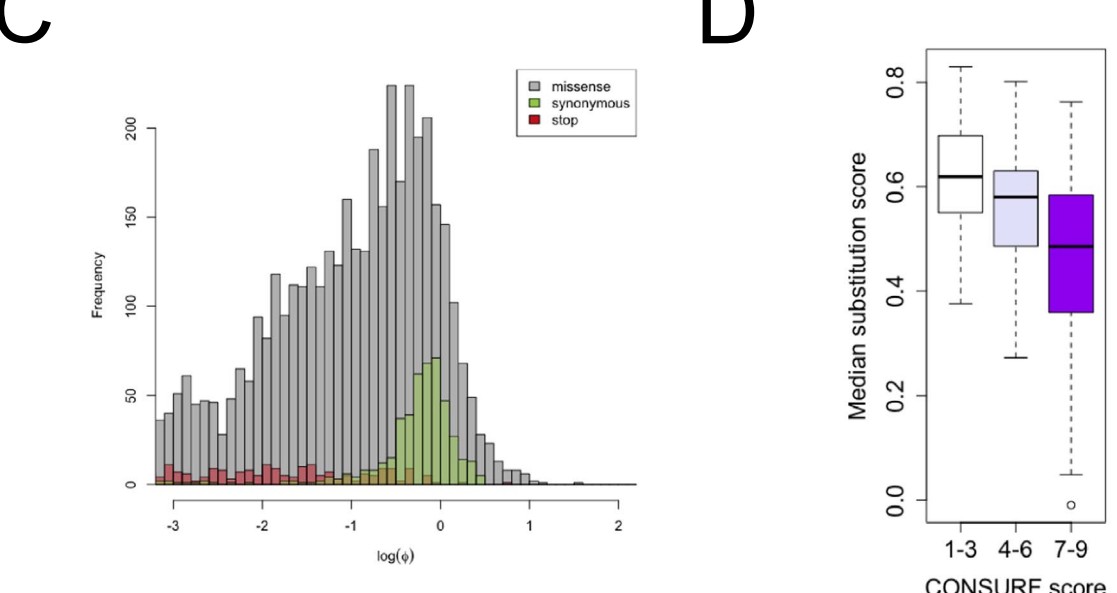

**Extended Data Fig. 1 | Massively parallel mutagenesis of *NOG2*. A.** Scheme for high throughput mutagenesis and analysis of functional variants of *NOG2*. **B.** Complete fitness landscape for variants in the core GTP-binding motifs of *NOG2* (G4, G1, G2, G3; amino acids 260-369). Yellow squares indicate the WT amino acid identity. **C.** Distributions of *observed* fitness scores (Φ) for nonsense (red), missense (gray) and synonymous (green) mutations. **D.** Correlation between per-residue median fitness score and ConSURF [41] score (evolutionary conservation: scores 1-3, low; scores 4-6, medium; scores 7-9, high conservation). The statistical analysis used was performed in TileSeq and described in Weile et al. (ref. [8]).

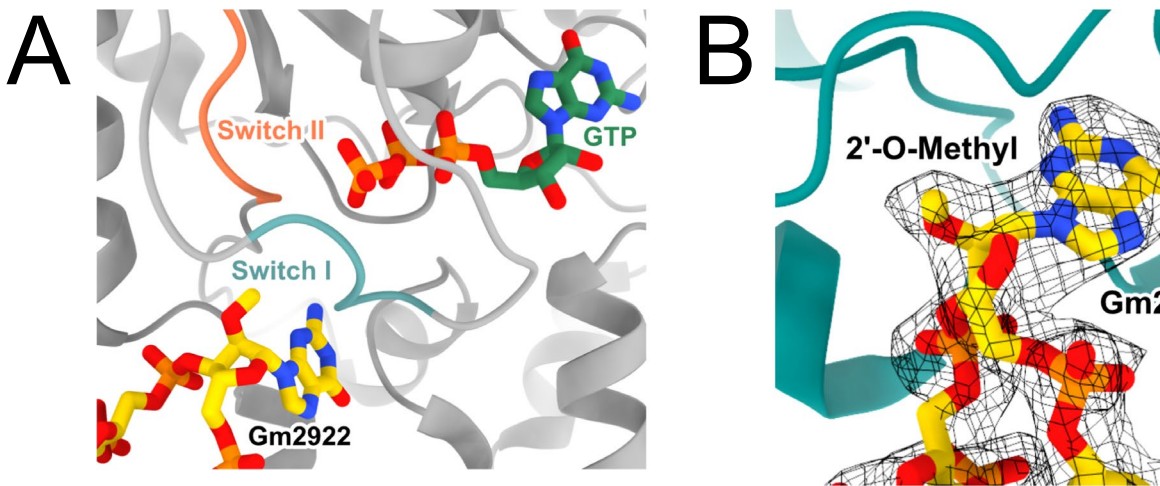

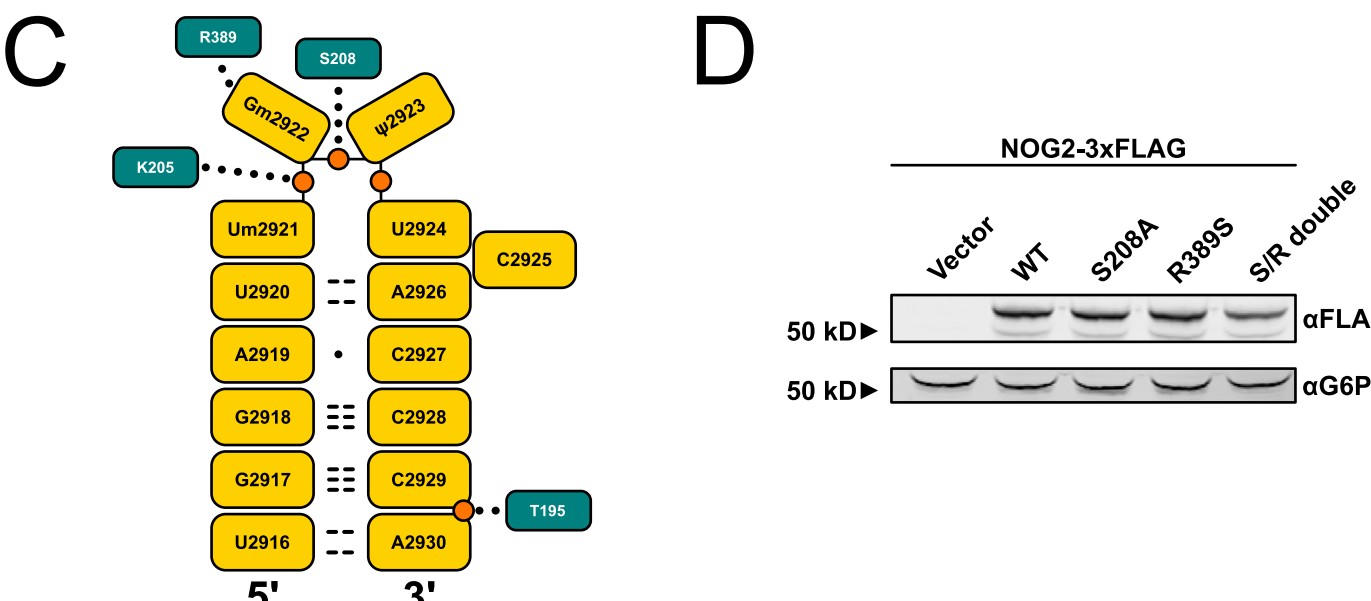

**Extended Data Fig. 2 | Details of interactions between Nog2 and H92/Gm2922. A**. Modeling of active site of Nog2 with Gm2922 (accession no. PDB 3JCT) shown and Switch I (G2) and Switch II (G3) colored. **B**. Cryo-EM map of Nog2 (EMDB-6615) showing clear density for 2′-O-methyl group at G2922. **C**. Two-dimensional map of interactions between Nog2 amino acids and RNA elements of H92. Nucleotides and amino acids are depicted as rectangles, relevant phosphates are shown as orange circles. Hydrogen bonds between bases are shown as dashed lines, and the wobble base pair between A2919 and C2927 is shown as a dot. Interactions between amino acids and H92 are depicted as dotted lines. Threonine 195 (T195) was selected as a candidate for a suppressing amino acid substitution (Fig. 3). **D**. Western blots to verify expression of WT NOG2-3xFLAG and variants S208A, R389S and double mutant. Glucose-6-phosphate dehydrogenase (G6PD) was used as a loading control.

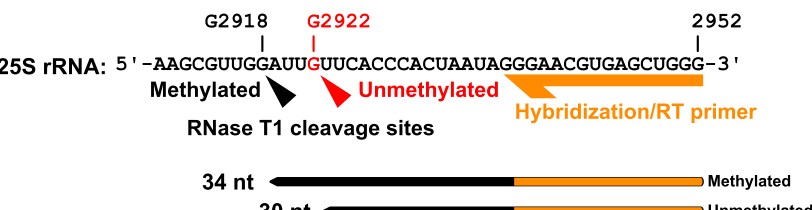

**Extended Data Fig. 3 | Schematic of primer extension probing 2'-O-methylation at G2922. A**. RNase T1 cleaves 3' to guanosines in single-stranded RNA. However, 2'-O-methylated guanosine is resistant to cleavage. Guanosines base-paired with the reverse transcription primer are also protected. After cleavage with RNase T1, reverse transcription will generate a 30 nt product if G2922 is sensitive to cleavage (that is not methylated), or a 34 nt product if G2922 is protected from cleavage (that is methylated).

# A

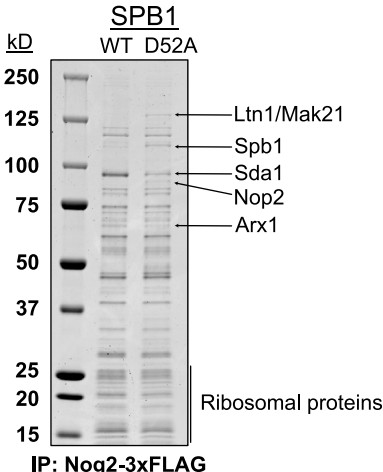

# B

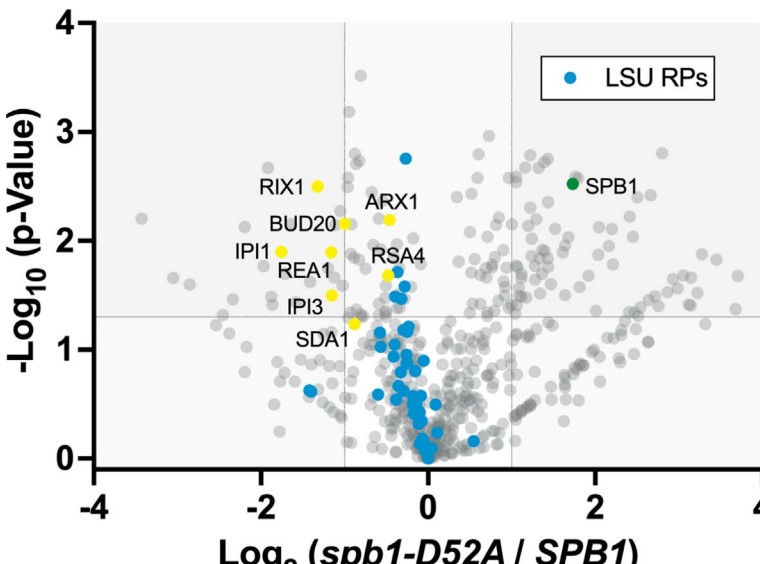

**Extended Data Fig. 4 | Nog2-3xFLAG immunopurifications and protein identification. A.** Immunopurifications from the indicated strains, stained with Coomassie blue. Proteins identified by mass spectrometry are indicated. **B.** Volcano plot showing log$_2$ fold change of factors that co-immunopurify with Nog2-3xFLAG, in unmodified G2922 (*spb1-DS2A*) versus modified Gm2922 (WT *SPB1*). Dashed lines intersecting x-axis indicate a log$_2$ fold change of 1, and intersecting y-axis indicate a -log$_{10}$ (*p*) value of 1.3 (*p* = 0.05).

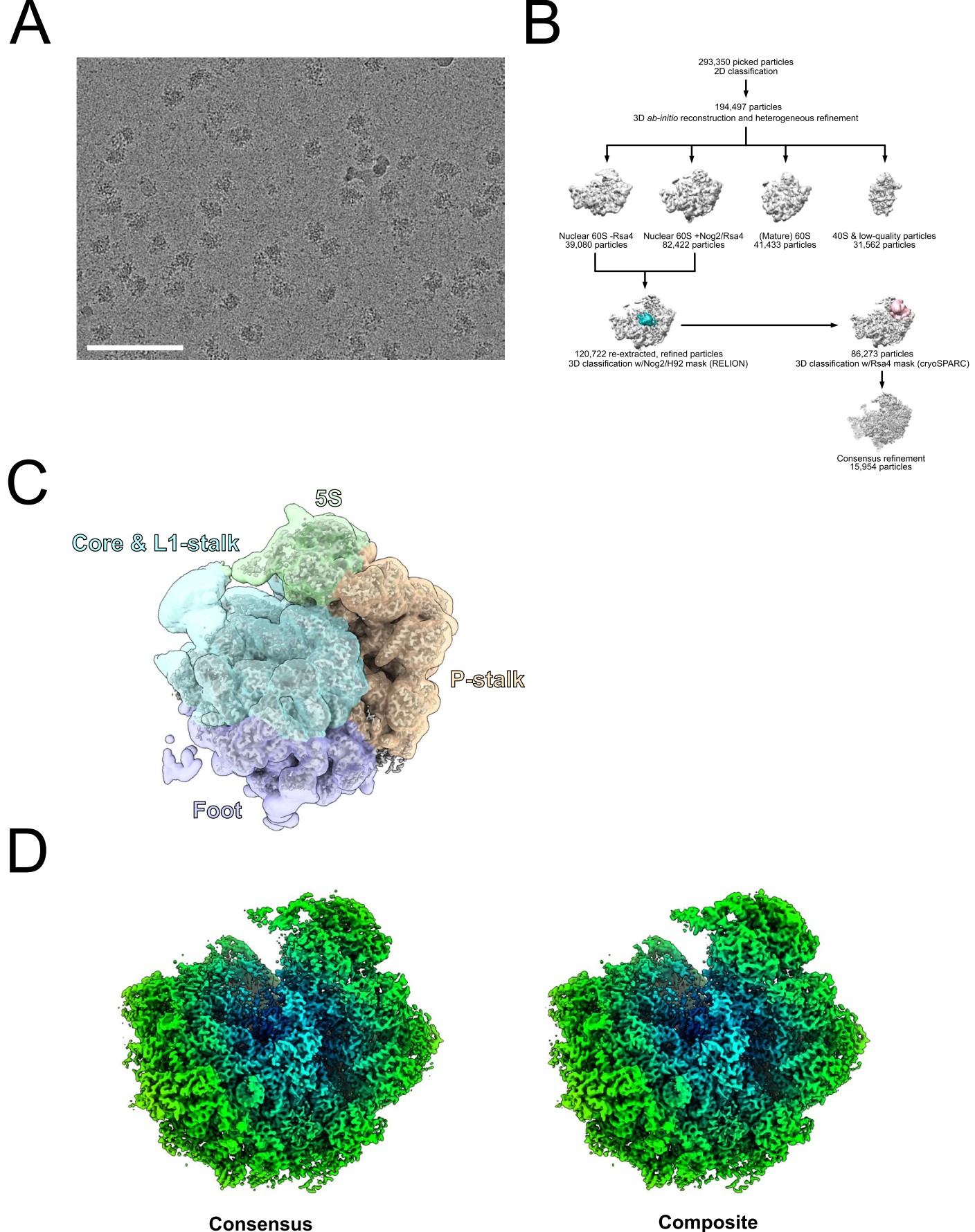

**Consensus**

**Composite**

**Extended Data Fig. 5 | Details of cryo-EM data collection and processing. A**. Raw micrograph of Nog2-3xFLAG particles immunopurified from the *pGAL:SPB1* strain AJY4425 expressing *spb1-D52A*. Representative of 25,956 micrographs. Scale bar = 100 nm. **B**. Cryo-EM data processing scheme by which the Nog2-3xFLAG structure was obtained.

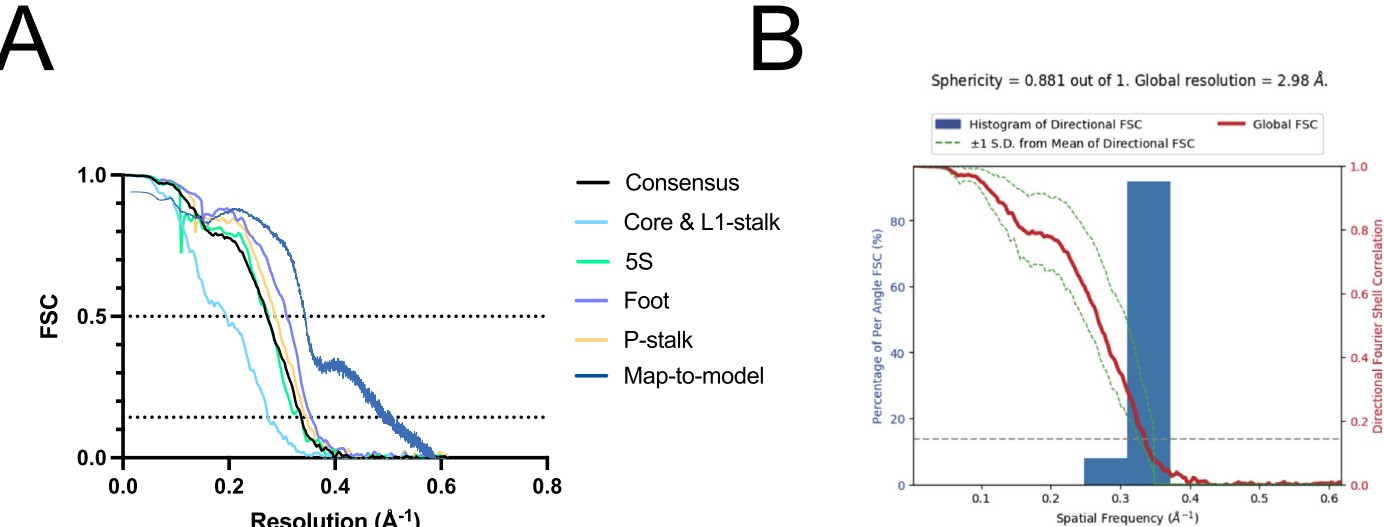

**Extended Data Fig. 6 | Cryo-EM data quality metrics. A**, **B**. Fourier shell correlation (FSC) curves **(A)**, and 655 distribution of viewing orientations **(B)** for the unmodified G2922 Nog2-3xFLAG structure. **C-D**. Directional **(C)** and map-to-model **(D)** FSC curves for the unmodified G2922 Nog2-3xFLAG structure. **E**. Local resolution analysis of the overall unmodified G2922 Nog2-3xFLAG structure.

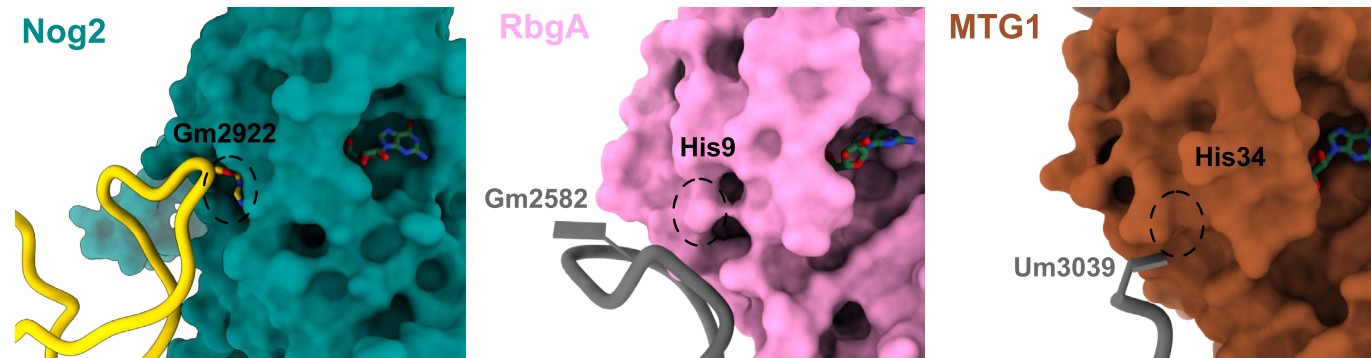

**Extended Data Fig. 7 | Details of the unmodified G2922 structure and structural comparison between Nog2 and orthologs. A**. Active site of Nog2 in complex with pre-60S lacking G2922 methylation. Densities for GDP, Mg²⁺ and ordered regions Switch I and Switch II are clearly observed. **B**. Cryo-EM map corresponding to H92 in the unmodified G2922 structure. **C**. Cartoon comparisons between H92 with a modified Gm2922 engaged with Nog2 (left, accession no. PDB 3JCT), an unmodified G2922 (middle, this work) and the mature H92 from the yeast ribosome crystal structure (right, accession no. PDB 4V88). **D**. Comparison between the active sites of Nog2 (left, accession no. PDB 3JCT), bacterial ribosome biogenesis GTPase RbgA (middle, accession no. PDB 6PPK) and human mitoribosome biogenesis GTPase MTG1 (right, accession no. PDB 7PD3). The position of Gm2922, or locations of histidine residues implicated in catalysis, are indicated by dashed circles.

# Reporting Summary

## Statistics

For all statistical analyses, confirm that the following items are present in the figure legend, table legend, main text, or Methods section.

| n/a | Confirmed | |
|---|---|---|
| ☐ | ☒ | The exact sample size (*n*) for each experimental group/condition, given as a discrete number and unit of measurement |
| ☐ | ☒ | A statement on whether measurements were taken from distinct samples or whether the same sample was measured repeatedly |
| ☐ | ☒ | The statistical test(s) used AND whether they are one- or two-sided *Only common tests should be described solely by name; describe more complex techniques in the Methods section.* |
| ☒ | ☐ | A description of all covariates tested |
| ☒ | ☐ | A description of any assumptions or corrections, such as tests of normality and adjustment for multiple comparisons |
| ☒ | ☐ | A full description of the statistical parameters including central tendency (e.g. means) or other basic estimates (e.g. regression coefficient) AND variation (e.g. standard deviation) or associated estimates of uncertainty (e.g. confidence intervals) |
| ☐ | ☒ | For null hypothesis testing, the test statistic (e.g. *F*, *t*, *r*) with confidence intervals, effect sizes, degrees of freedom and *P* value noted *Give P values as exact values whenever suitable.* |
| ☒ | ☐ | For Bayesian analysis, information on the choice of priors and Markov chain Monte Carlo settings |
| ☒ | ☐ | For hierarchical and complex designs, identification of the appropriate level for tests and full reporting of outcomes |
| ☒ | ☐ | Estimates of effect sizes (e.g. Cohen's *d*, Pearson's *r*), indicating how they were calculated |

*Our web collection on statistics for biologists contains articles on many of the points above.*

## Software and code

Policy information about availability of computer code

| Data collection | Data were collected on an FEI Titan Krios microscope with a K3 direct electron detector using Serial EM v3.8. |
|---|---|
| Data analysis | Cyber T webserver was used to analyze mass spectrometry data. Cryo-EM data was analyzed using cryoSPARC (Live and v3.2) and RELION (v3.1.3). The model was built using COOT v1.0 and relaxed using flexible molecular dynamics fitting with ISOLDE v1.4, then finally subjected to real-space refinement as implemented in PHENIX v1.19.2. All molecular graphics were prepared with UCSF ChimeraX v1.0. |

For manuscripts utilizing custom algorithms or software that are central to the research but not yet described in published literature, software must be made available to editors and reviewers. We strongly encourage code deposition in a community repository (e.g. GitHub). See the Nature Portfolio guidelines for submitting code & software for further information.

## Data

Policy information about availability of data

All manuscripts must include a data availability statement. This statement should provide the following information, where applicable:
- Accession codes, unique identifiers, or web links for publicly available datasets
- A description of any restrictions on data availability
- For clinical datasets or third party data, please ensure that the statement adheres to our policy

The raw sequencing data are deposited in the Sequencing Read Archive (SRA) with accession number PRJNA892951. The structure of the Nog2-3xFLAG unmodified pre-60S subunit and its associated atomic coordinates have been deposited into the Electron Microscopy Data Bank (EMDB) and the Protein Data Bank (PDB) as EMDB-26485, and PDB 7UG6, respectively. All other data are available in the manuscript or supplementary materials. Requests for strains or plasmids will be fulfilled by the lead contact author, AWJ, upon request.

# Field-specific reporting

Please select the one below that is the best fit for your research. If you are not sure, read the appropriate sections before making your selection.

☒ Life sciences        ☐ Behavioural & social sciences        ☐ Ecological, evolutionary & environmental sciences

For a reference copy of the document with all sections, see nature.com/documents/nr-reporting-summary-flat.pdf

# Life sciences study design

All studies must disclose on these points even when the disclosure is negative.

| | |
|---|---|
| Sample size | The number of particles used were limited by the amount of time that data could be collected. |
| Data exclusions | No data was excluded. |
| Replication | Genetic screens, biochemical assays, and cell imaging are representative of two independent experiments. Multiple structures were obtained at various time points that resembled one another. |
| Randomization | No randomization was performed. Randomization was not relevant to this study because it didnot involve human subjects or live animals where bias is possible. |
| Blinding | No blinding was performed. Blinding is not relevant to this study because it did not involve human subjects or liver animals where bias is possible. |

# Reporting for specific materials, systems and methods

We require information from authors about some types of materials, experimental systems and methods used in many studies. Here, indicate whether each material, system or method listed is relevant to your study. If you are not sure if a list item applies to your research, read the appropriate section before selecting a response.

### Materials & experimental systems

| n/a | Involved in the study |
|---|---|
| ☐ | ☒ Antibodies |
| ☒ | ☐ Eukaryotic cell lines |
| ☒ | ☐ Palaeontology and archaeology |
| ☒ | ☐ Animals and other organisms |
| ☒ | ☐ Human research participants |
| ☒ | ☐ Clinical data |
| ☒ | ☐ Dual use research of concern |

### Methods

| n/a | Involved in the study |
|---|---|
| ☒ | ☐ ChIP-seq |
| ☒ | ☐ Flow cytometry |
| ☒ | ☐ MRI-based neuroimaging |

## Antibodies

| | |
|---|---|
| Antibodies used | Primary antibodies used included rabbit anti-G6PD and rat anti-FLAG (Agilent). Secondary antibodies included  goat anti-rabbit 680RD and goat anti-rat 800CW. |
| Validation | Primary antibodies have been verified by relative expression and knockdown to confirm specificity. |

