## [Peer Review File · Nature Structural & Molecular Biology]

Peer Review Information

Manuscript Title: A single 2'-O-methylation of ribosomal RNA gates assembly of a functional ribosome

Corresponding author name(s): David W. Taylor, Arlen W. Johnson

Reviewer Comments & Decisions:

Decision Letter, presubmission inquiry:
--

Message:

Nature Structural & Molecular Biology NSMB-PI46027

17th Feb 2022

Dear Dr. Taylor,

Thank you for the presubmission inquiry regarding your manuscript "A single 2'-O-methylation of ribosomal RNA gates assembly of a functional ribosome". Based on the brief summary of your study, we are interested in considering your report for possible publication in Nature Structural & Molecular Biology. Please note that it is difficult to accurately assess a manuscript based on a brief description. Therefore, while we look forward to receiving your paper, we cannot guarantee that we would necessarily send the report out for review until we have read the entire manuscript.

In order to submit your complete manuscript, please use the link below:

[Redacted]

Sincerely,
Sara

Sara Osman, Ph.D.
Associate Editor
Nature Structural & Molecular Biology

Decision Letter, initial submission:

Message: 25th May 2022

Dear Dr. Taylor,

Thank you again for submitting your manuscript "A single 2'-O-methylation of ribosomal RNA gates assembly of a functional ribosome". I sincerely apologize for the delay in responding, which resulted from the difficulty in obtaining suitable referee reports. Nevertheless, we now have comments (below) from the 2 reviewers who evaluated your paper. In light of those reports, we remain interested in your study and would like to see your response to the comments of the referees, in the form of a revised manuscript.

Please be sure to address/respond to all concerns of the referees in full in a point-by-point response and highlight all changes in the revised manuscript text file. If you have comments that are intended for editors only, please include those in a separate cover letter.

You will see that reviewer #2 suggests addition of functional validation of the observed role of the rRNA methylation in ribosome biogenesis - while we agree that these experiments would increase the impact of your study, please reach out if you wish to discuss how to address this point.

We expect to see your revised manuscript within 6 weeks. If you cannot send it within this time, please contact us to discuss an extension; we would still consider your revision, provided that no similar work has been accepted for publication at NSMB or published elsewhere.

Reporting Summary:

When submitting the revised version of your manuscript, please pay close attention to our [href="https://www.nature.com/nature-research/editorial-policies/image-integrity">Digital Image Integrity Guidelines. and to the following points below:](https://www.nature.com/nature-research/editorial-policies/image-integrity)

-- that unprocessed scans are clearly labelled and match the gels and western blots

presented in figures.

-- that control panels for gels and western blots are appropriately described as loading on sample processing controls

-- all images in the paper are checked for duplication of panels and for splicing of gel lanes.

Please note that all key data shown in the main figures as cropped gels or blots should be presented in uncropped form, with molecular weight markers. These data can be aggregated into a single supplementary figure item. While these data can be displayed in a relatively informal style, they must refer back to the relevant figures. These data should be submitted with the final revision, as source data, prior to acceptance, but you may want to start putting it together at this point.

Data availability: this journal strongly supports public availability of data. All data used in accepted papers should be available via a public data repository, or alternatively, as Supplementary Information. If data can only be shared on request, please explain why in your Data Availability Statement, and also in the correspondence with your editor. Please note that for some data types, deposition in a public repository is mandatory - more information on our data deposition policies and available repositories can be found below: <https://www.nature.com/nature-research/editorial-policies/reporting-standards#availability-of-data>

We require deposition of coordinates (and, in the case of crystal structures, structure factors) into the Protein Data Bank with the designation of immediate release upon publication (HPUB). Electron microscopy-derived density maps and coordinate data must be deposited in EMDb and released upon publication. Deposition and immediate release of NMR chemical shift assignments are highly encouraged. Deposition of deep sequencing and microarray data is mandatory, and the datasets must be released prior to or upon publication. To avoid delays in publication, dataset accession numbers must be supplied with the final accepted manuscript and appropriate release dates must be indicated at the galley proof stage.

[Redacted]

Sincerely,

Sara Osman, Ph.D.
Associate Editor
Nature Structural & Molecular Biology

Referee expertise:

Referee #1: Cryo-EM, RNA modifications

Referee #2: Ribosome biogenesis, yeast genetics

Reviewers' Comments:

Reviewer #1:
Remarks to the Author:

Ribosome assembly is an essential and complex cellular process. One area of ribosome assembly that has been poorly studied is the role of rRNA modifications. In this manuscript the authors used a scanning mutagenesis approach to identify important regions of the GTPase Nog2, which is a critical ribosome assembly factor. From this screen the authors discovered that Nog2 interacts with the 2'-O-methylated A-site base Gm2922, via residues R389 and S208. This association was confirmed by creating a Nog2

S208A/R389S double variant which is lethal in yeast. G2922 is specifically methylated by the MTase Spb1. To test the significance of this modification the authors used an artificial snoRNA to guide methylation in an Spd1 deficient strain. Through this approach they found that methylation of G2922 is essential for maturation of the LSU and nuclear export of nascent-LSU particles. To test the hypothesis that Nog1 monitors this methylation the authors created a Nog2 mutant with enhanced binding to H92 from the LSU. This mutant was able to overcome the growth defect in Spb1 deficient cells. Next the authors isolated pre-LSU particles from Spb1 deficient cells and found that loss of methylation leads to the accumulation of nucleolar assembly factors. Finally, the authors determined the cryo-EM structure of an unmodified pre-60S particle bound by Nog2 revealing that without methylation G2922 is an altered conformation and not present in the active site channel of Nog2. Through the combination of scanning mutagenesis, yeast genetics, cell-based assays, mass-spec, and cryo-EM this manuscript establishes that methylation of G2922 is a crucial structural checkpoint in ribosome assembly. I commend the authors on the discovery of the significance of this rRNA modification! I fully support publication of this manuscript in NSMB.

Minor Concerns:

- The mutagenesis tolerance scale in Fig. 1A and 1C is missing or not visible.
- Can the authors confirm that the Nog2 mutants shown in Fig. 1E express in yeast?
- Aside from the structural changes in H92 are there any other differences in the cryo-EM structure of the unmodified LSU particle?

Reviewer #2:

Remarks to the Author:

Yelland et al.

This manuscript combines genetic data and cryo-EM to demonstrate that methylation of G2922 in 25S rRNA is an essential checkpoint for ribosome assembly, inspected by the Nog2 GTPase. The work is carefully executed, and very interesting. However, what is missing is a logic for the observation. What is the role for this methylation? These experiments should, in my opinion, be part of a high impact publication as in Nature Structural and Molecular Biology.

Major issues:

1. The cryo-EM Figures are difficult to evaluate as they do not contain local density. The authors should include that either in the main Figure 4B&C or in a supplemental panel.
2. The manuscript shows the importance of the modification for assembly. However, the manuscript also shows that it can be bypassed without large defects in growth. The authors should test if the resulting ribosomes from the Spb1_D52A/Nog2T195R or H392R strain have functional defects.
3. P.4, l.165: what is the data supporting the statement that Spb1 assesses the correct folding of the proto A-site?
4. The authors might want to speculate (once they have some functional data) why this modification is so critical.

Minor issues:

1. Abstract l.27, please insert the word "can". Not all modifications are likely to have such

a role

2. Figure 1A,B and S6E lack the color color gradient that functions as a legend.

3. What are the yellow squares in Figure S1B?

4. What are the other significantly changed proteins in Figure S4? Why are they not discussed?

5. what happened to all the particles that didnt make the cut for the final model? are they different? are they damaged? are they just worse resolution? if so why? the authors might want to add a couple of sentences in the methods

Author Rebuttal to Initial comments

Response to Reviewers

We appreciate the enthusiasm of the reviewers. They “commend... the discovery of the significance of this RNA modification!” We also value their constructive comments. We have done our best to address these, and we believe that addressing these points has significantly strengthened the manuscript.

Reviewer #1:

Remarks to the Author:

Ribosome assembly is an essential and complex cellular process. One area of ribosome assembly that has been poorly studied is the role of rRNA modifications. In this manuscript the authors used a scanning mutagenesis approach to identify important regions of the GTPase Nog2, which is a critical ribosome assembly factor. From this screen the authors discovered that Nog2 interacts with the 2'-O-methylated A-site base Gm2922, via residues R389 and S208. This association was confirmed by creating a Nog2 S208A/R389S double variant which is lethal in yeast. G2922 is specifically methylated by the MTase Spb1. To test the significance of this modification the authors used an artificial snoRNA to guide methylation in an Spd1 deficient strain. Through this approach they found that methylation of G2922 is essential for maturation of the LSU and nuclear export of nascent-LSU particles. To test the hypothesis that Nog1 monitors this methylation the authors created a Nog2 mutant with enhanced binding to H92 from the LSU. This mutant was able to overcome the growth defect in Spb1 deficient cells. Next the authors isolated pre-LSU particles from Spb1 deficient cells and found that loss of methylation leads to the accumulation of nucleolar assembly factors. Finally, the authors determined the cryo-EM structure of an unmodified pre-60S particle bound by Nog2 revealing that without methylation G2922 is in an altered conformation and not present in the active site channel of Nog2. Through the combination of scanning mutagenesis, yeast genetics, cell-based assays, mass-spec, and cryo-EM this manuscript establishes that methylation of G2922 is a crucial structural checkpoint in ribosome assembly. I commend the authors on the discovery of the significance of this rRNA modification! I fully support publication of this manuscript in NSMB.

We thank the reviewer for their overwhelmingly enthusiastic review.

Minor Concerns:

- The mutagenesis tolerance scale in Fig. 1A and 1C is missing or not visible.

We thank the reviewer for pointing out this omission, which happened during PDF conversion. It has been corrected.

- Can the authors confirm that the Nog2 mutants shown in Fig. 1E express in yeast?

This is an excellent point. Using a Western blot to probe for 3xFLAG-tagged Nog2, we now show that all three Nog2 mutants are expressed at similar levels in yeast. A new Supplementary Figure (Supplementary Fig. S2D) shows these data.

- Aside from the structural changes in H92 are there any other differences in the cryo-EM structure of the unmodified LSU particle?

This is a great question. Compared to model 3JCT, a similar immunopurification of Nog2 in complex with the (wild-type) nascent 60S, we observe relatively few changes in the structure of the unmodified LSU. In contrast to model 3JCT, we observe that ITS2 has been processed, Rpl12 has been loaded and that Alb1 and YBL028Cp are present, as observed in later nucleoplasmic complexes (e.g. PDB 6YLH, reference 12). Furthermore, we note that Arx1 and Bud20 are present but apparently substoichiometric, which may result from nucleolar retention of the unmodified particle. Finally, we note that Nog2 appears to be bound to GDP instead of GTP. Importantly, we do not observe the presence of unknown protein factors; nor, with the exception of H92, significant deviations from previously observed states of ribosomal RNA. These observations support our conclusion that lack of Gm2922 has a very specific effect on the stability and/or activity of Nog2.

Reviewer #2:

Remarks to the Author:

Yelland et al.

This manuscript combines genetic data and cryo-EM to demonstrate that methylation of G2922 in 25S rRNA is an essential checkpoint for ribosome assembly, inspected by the Nog2 GTPase. The work is carefully executed, and very interesting. However, what is missing is a logic for the observation. What is the role for this methylation? These experiments should, in my opinion, be part of a high impact publication as in Nature Structural and Molecular Biology.

We thank the reviewer for finding the manuscript “carefully executed and very interesting.”

Major issues:

1. The cryo-EM Figures are difficult to evaluate as they do not contain local density. The authors should include that either in the main Figure 4B&C or in a supplemental panel.

We thank the reviewer for pointing out this important oversight. We now include local density for Nog2-bound nucleotide and H92 in new Supplementary Figures in the revision (Supplementary Figs. S7A and S7B).

2. The manuscript shows the importance of the modification for assembly. However, the manuscript also shows that it can be bypassed without large defects in growth. The authors should test if the resulting ribosomes from the Spb1_D52A/Nog2T195R or H392R strain have functional defects.

We agree with the reviewer that this is an important question. However, we believe that addressing this point merits a much more comprehensive analysis than what the reviewer asks for. In our current story, we set out to investigate the role of this single RNA modification in ribosome assembly, and the reviewer is convinced that we have done so. Experiments addressing the function of the modification in translation will be done as part of a larger story and will be the basis for a future manuscript, precisely because they are extremely interesting. In that work, we will determine the function of the modified RNA residue that we examined in our current work, as well as that of two adjacent residues, on translation. We will examine the interdependence of these modifications, their effects on translation and cell fitness in vivo and will use in vitro biochemical analyses to rigorously determine their impact on rates of peptidyl transferase activity. This work will require establishing collaborations with at least two other labs. We believe that this comprehensive analysis will result in at least one more highly interesting story, but we hope the reviewer shares our thinking that these experiments are beyond the scope of the current manuscript.

3. P.4, l.165: what is the data supporting the statement that Spb1 assesses the correct folding of the proto A-site?

This statement is meant to convey that Spb1-catalyzed methylation of G2922 is a precursor to maturation of the A-site. We have changed the text to "Spb1 prepares the RNA of the proto A-site for binding by Nog2" to be more accurate, and we appreciate the reviewer's clarifying question.

4. The authors might want to speculate (once they have some functional data) why this modification is so critical.

We have now included a few ideas about the possibility that this modification impacts ribosome function in the Discussion section of the manuscript.

Minor issues:

1. Abstract l.27, please insert the word "can". Not all modifications are likely to have such a role

This modification has been added.

2. Figure 1A,B and S6E lack the color color gradient that functions as a legend.

We apologize for this omission, which seemed to happen during pdf conversion. This has been corrected.

3. What are the yellow squares in Figure S1B?

We thank the reviewer for pointing out this oversight. The yellow squares correspond to WT amino acid identities, and this information has been added to the legend of Figure S1B.

4. What are the other significantly changed proteins in Figure S4? Why are they not discussed?

We appreciate the reviewer's insightful question. Since a lack of Gm2922 inhibits nuclear export of the pre-60S, "upstream" steps of the ribosome biogenesis pathway are also stalled by the greatly decreased flux of particles. In general, peptide counts that are increased with respect to the modified subunit correspond with earlier nucleolar steps of 60S biogenesis, presumably because later steps are inhibited. We feel that in the context of this manuscript, the sheer number of nucleolar factors involved in ribosome biogenesis preclude meaningful discussion, despite the fascinating potential for deeper discussion. We do suspect that Nog2 is capable of binding the pre-60S before its H92 binding site is fully established, but under normal circumstances, does not do so until H92 is modified and clear of Spb1. We also provide all MS peptide counts in the supplemental data for use by other researchers in the field.

5. what happened to all the particles that didnt make the cut for the final model? are they different? are they damaged? are they just worse resolution? if so why? the authors might want to add a couple of sentences in the methods

This is an excellent question. We have included information in the Methods section addressing this point.

"Following on-the-fly 2D classification to separate clean pre-60S projections from unassignable or "junk" classes, particles were exported to cryoSPARC⁴⁴ (version 3.2), where multiple subsequent rounds of 2D classification resulted in a total of 194,497 particles. These particles were used for *ab initio* reconstruction and 3D heterogeneous refinement to separate a total of 120,722 nucl(eol)ar particles with at least partial Nog2 occupancy, which were subjected to consensus non-uniform refinement. Using the `csparc2star.py` function in pyEM⁴⁵, these particles were exported to RELION⁴⁶ (version 3.1.3), where a 3D classification scheme using a soft mask around Nog2/H92 was used to further separate 86,273 Nog2-bound particles. These particles were re-imported into cryoSPARC for a 3D classification scheme with a soft mask around Rsa4, which separated 15,954 particles with Nog2 at high resolution. Separation of Rsa4-bound Nog2 particles was key to obtaining the highest-possible resolution map of both Nog2 and the 5S particle, which is flexible prior to Rea1-mediated rotation. The final map was

prepared using non-uniform refinement implemented in cryoSPARC, where FSC calculations and local resolution analysis were also carried out after final refinement.”

Decision Letter, first revision:

Message: Our ref: NSMB-A46027B

28th Jul 2022

Dear Dr. Taylor,

Thank you for submitting your revised manuscript "A single 2'-O-methylation of ribosomal RNA gates assembly of a functional ribosome" (NSMB-A46027B). We have assessed the revised manuscript in-house and found that the paper has improved in revision, and therefore we'll be happy in principle to publish it in Nature Structural & Molecular Biology, pending minor revisions to satisfy any remaining referees' final requests and to comply with our editorial and formatting guidelines.

We are now performing detailed checks on your paper and will send you a checklist detailing our editorial and formatting requirements in about two weeks. Please do not upload the final materials and make any revisions until you receive this additional information from us.

To facilitate our work at this stage, we would appreciate if you could send us the main text as a word file. Please make sure to copy the NSMB account (cc'ed above).

Sincerely,
Sara

Sara Osman, Ph.D.
Associate Editor
Nature Structural & Molecular Biology

Decision letter, author guidance

Message: Our ref: NSMB-A46027B

12th Oct 2022

Dear Dr. Taylor,

Thank you for your patience as we've prepared the guidelines for final submission of your Nature Structural & Molecular Biology manuscript, "A single 2'-O-methylation of ribosomal

RNA gates assembly of a functional ribosome" (NSMB-A46027B). Please carefully follow the step-by-step instructions provided in the attached file, and add a response in each row of the table to indicate the changes that you have made. Please also check and comment on any additional marked-up edits we have proposed within the text. Ensuring that each point is addressed will help to ensure that your revised manuscript can be swiftly handed over to our production team.

We would like to start working on your revised paper, with all of the requested files and forms, as soon as possible. If you can resubmit within the next week it is possible that your submission could be published before the end of 2022. Please get in contact with us if you anticipate any delays in resubmission.

In recognition of the time and expertise our reviewers provide to Nature Structural & Molecular Biology's editorial process, we would like to formally acknowledge their contribution to the external peer review of your manuscript entitled "A single 2'-O-methylation of ribosomal RNA gates assembly of a functional ribosome". For those reviewers who give their assent, we will be publishing their names alongside the published article.

Nature Structural & Molecular Biology offers a Transparent Peer Review option for new original research manuscripts submitted after December 1st, 2019. As part of this initiative, we encourage our authors to support increased transparency into the peer review process by agreeing to have the reviewer comments, author rebuttal letters, and editorial decision letters published as a Supplementary item. When you submit your final files please clearly state in your cover letter whether or not you would like to participate in this initiative. Please note that failure to state your preference will result in delays in accepting your manuscript for publication.

Cover suggestions

As you prepare your final files we encourage you to consider whether you have any images or illustrations that may be appropriate for use on the cover of Nature Structural & Molecular Biology.

Nature Structural & Molecular Biology has now transitioned to a unified Rights Collection system which will allow our Author Services team to quickly and easily collect the rights and permissions required to publish your work. Approximately 10 days after your paper is formally accepted, you will receive an email in providing you with a link to complete the grant of rights. If your paper is eligible for Open Access, our Author Services team will also be in touch regarding any additional information that may be required to arrange payment for your article.

Please note that *Nature Structural & Molecular Biology* is a Transformative Journal (TJ). Authors may publish their research with us through the traditional subscription access route or make their paper immediately open access through payment of an article-processing charge (APC). Authors will not be required to make a final decision about access to their article until it has been accepted. [Find out more about Transformative Journals](https://www.springernature.com/gp/open-research/transformative-journals)

Authors may need to take specific actions to achieve [compliance with funder and institutional open access mandates](https://www.springernature.com/gp/open-research/funding/policy-compliance-faqs). If your research is supported by a funder that requires immediate open access (e.g. according to [Plan S principles](https://www.springernature.com/gp/open-research/plan-s-compliance)) then you should select the gold OA route, and we will direct you to the compliant route where possible. For authors selecting the subscription publication route, the journal's standard licensing terms will need to be accepted, including [self-archiving policies](https://www.nature.com/nature-portfolio/editorial-policies/self-archiving-and-license-to-publish). Those licensing terms will supersede any other terms that the author or any third party may assert apply to any version of the manuscript.

Please use the following link for uploading these materials:
[Redacted]

Best regards,

Sophia Frank
Editorial Assistant
Nature Structural & Molecular Biology
nsmb@us.nature.com

On behalf of

Sara Osman, Ph.D.
Associate Editor
Nature Structural & Molecular Biology

Author Rebuttal, first revision

There are no comments from reviewers for a response.

Final Decision Letter:

Message 4th Nov 2022

:

Dear Dr. Taylor,

We are now happy to accept your revised paper "A single 2'-O-methylation of ribosomal RNA gates assembly of a functional ribosome" for publication as a Article in Nature Structural & Molecular Biology.

Due to the importance of these deadlines, we ask that you please let us know now whether you will be difficult to contact over the next month. If this is the case, we ask you provide

us with the contact information (email, phone and fax) of someone who will be able to check the proofs on your behalf, and who will be available to address any last-minute problems.

As soon as your article is published, you can generate your shareable link by entering the DOI of your article here: http://authors.springernature.com/share. Corresponding authors will also receive an automated email with the shareable link

Your paper will be published online soon after we receive proof corrections and will appear in print in the next available issue. You can find out your date of online publication by contacting the production team shortly after sending your proof corrections. Content is published online weekly on Mondays and Thursdays, and the embargo is set at 16:00 London time (GMT)/11:00 am US Eastern time (EST) on the day of publication. Now is the time to inform your Public Relations or Press Office about your paper, as they might be interested in promoting its publication. This will allow them time to prepare an accurate and satisfactory press release. Include your manuscript tracking number (NSMB-A46027C) and our journal name, which they will need when they contact our press office.

About one week before your paper is published online, we shall be distributing a press release to news organizations worldwide, which may very well include details of your work. We are happy for your institution or funding agency to prepare its own press release, but it must mention the embargo date and Nature Structural & Molecular Biology. If you or your Press Office have any enquiries in the meantime, please contact press@nature.com.

Please note that *Nature Structural & Molecular Biology* is a Transformative Journal (TJ). Authors may publish their research with us through the traditional subscription access route or make their paper immediately open access through payment of an article-processing charge (APC). Authors will not be required to make a final decision about access to their article until it has been accepted. Find out more about Transformative Journals

Authors may need to take specific actions to achieve compliance with funder and institutional open access mandates. If your research is supported by a funder that requires immediate open access (e.g. according to Plan S principles) then you should select the gold OA route, and we will direct you to the compliant route where possible. For authors selecting the subscription publication route, the journal's standard licensing terms will need to be accepted, including self-archiving policies. Those licensing terms will supersede any other terms that the author or any third party may assert apply to any version of the manuscript.

Sincerely,
Sara

Sara Osman, Ph.D.
Associate Editor
Nature Structural & Molecular Biology
